# A biodegradable hybrid inorganic nanoscaffold for advanced stem cell therapy

Letao Yang [1], Sy-Tsong Dean Chueng[1], Ying Li[2], Misaal Patel[2], Christopher Rathnam[1], Gangotri Dey[1], Lu Wang[1], Li Cai[2] & Ki-Bum Lee[1,2,3]

Stem cell transplantation, as a promising treatment for central nervous system (CNS) diseases, has been hampered by crucial issues such as a low cell survival rate, incomplete differentiation, and limited neurite outgrowth in vivo. Addressing these hurdles, scientists have designed bioscaffolds that mimic the natural tissue microenvironment to deliver physical and soluble cues. However, several significant obstacles including burst release of drugs, insufficient cellular adhesion support, and slow scaffold degradation rate remain to be overcome before the full potential of bioscaffold–based stem-cell therapies can be realized. To this end, we developed a biodegradable nanoscaffold-based method for enhanced stem cell transplantation, differentiation, and drug delivery. These findings collectively support the therapeutic potential of our biodegradable hybrid inorganic (BHI) nanoscaffolds for advanced stem cell transplantation and neural tissue engineering.

[1] Department of Chemistry and Chemical Biology, Rutgers, The State University of New Jersey, 123 Bevier Road, Piscataway, NJ 08854, USA. [2] Department of Biomedical Engineering, Rutgers, The State University of New Jersey, 599 Taylor Road, Piscataway, NJ 08854, USA. [3] College of Pharmacy, Kyung Hee University, 26 Kyungheedae-ro, Dongdaemun-gu, Seoul 02447, Korea. Correspondence and requests for materials should be addressed to K.-B.L. (email: kblee@rutgers.edu)

Developing reliable therapeutic methods to treat central nervous system (CNS) diseases (e.g., Alzheimer's and Parkinson's diseases), degeneration in the aging brain, and CNS injuries (e.g., spinal cord injury (SCI) and traumatic brain injuries) has been a major challenge due to the complex and dynamic cellular microenvironment during the disease progression[1,2]. Several current therapeutic approaches have aimed to restore neural signaling, reduce neuroinflammation, and prevent subsequent damage to the injured area using stem cell transplantations[3–6]. Given the intrinsically limited regenerative abilities of the CNS and the highly complex inhibitory environment of the damaged tissues, stem cell transplantation has great potential to regenerate a robust population of functional neural cells such as neurons and oligodendrocytes, thereby re-establishing disrupted neural circuits in the damaged CNS areas[4,7–10]. However, several pertinent obstacles hinder advances in stem cell transplantation. First, due to the inflammatory nature of the injured regions, many transplanted cells perish soon after transplantation[11]. Second, the extracellular matrix (ECM) of the damaged areas is not conducive to stem cell survival and differentiation[2,12]. Therefore, to address the aforementioned issues and facilitate the progress of stem cell therapies, there is a clear need to develop an innovative approach to increase the survival rate of transplanted stem cells and to better control stem cell fate in vivo, which can lead to the recovery of the damaged neural functions and the repair of neuronal connections in a more effective manner.

To this end, we report a biodegradable hybrid inorganic (BHI) nanoscaffold-based method to improve the transplantation of human patient-derived neural stem cells (NSCs) and to control the differentiation of transplanted NSCs in a highly selective and efficient way. Further, as a proof-of-concept demonstration, we combined the spatiotemporal delivery of therapeutic molecules with enhanced stem cell survival and differentiation using BHI-nanoscaffold in a mouse model of SCI. Specifically, our developed three-dimensional (3D) BHI-nanoscaffolds (Fig. 1) have unique benefits for advanced stem cell therapies: (i) wide-range tunable biodegradation; (ii) upregulated ECM-protein binding affinity; (iii) highly efficient drug loading with sustained drug delivery capability; and (iv) innovative magnetic resonance imaging (MRI)-based drug release monitoring (Fig. 1a-c). Hybrid biomaterial scaffolds have been demonstrated to mimic the natural microenvironment for stem cell-based tissue engineering[13–22]. In this regard, scientists including our group, have recently reported that low-dimensional (0D, 1D, and 2D) inorganic and carbon nanomaterial (e.g., TiO2 nanotubes, carbon nanotubes, and graphene)-based scaffolds, having unique biological and physiochemical properties, and nanotopographies, can effectively control stem cell behaviors in vitro, as well as in vivo[23–31]. However, these inorganic and carbon-based nanoscaffolds are intrinsically limited by their non-biodegradability and restricted biocompatibility, thereby delaying their wide clinical applications. On the contrary, $MnO_2$ nanomaterials have proven to be biodegradable in other bioapplications such as cancer therapies, with MRI active $Mn^{2+}$ ions as a degradation product[32–34]. Taking advantage of their biodegradability, and incorporating their unique physiochemical properties into stem cell-based tissue engineering, we have developed $MnO_2$ nanomaterials-based 3D hybrid nanoscaffolds to better regulate stem cell adhesion, differentiation into neurons, and neurite outgrowth in vitro and for enhanced stem cell transplantation in vivo (Fig. 1d-e). Considering the difficulties of generating a robust population of functional neurons and enhancing neuronal behaviors (neurite outgrowth and axon regeneration), our biodegradable $MnO_2$ nanoscaffold can potentially serve as a powerful tool for improving stem cell transplantation and advancing stem cell therapy.

## Results

### Enhanced stem cell differentiation on $MnO_2$ nanoscaffolds.

Recently, hybrid inorganic 2D nanomaterial-based scaffolds have been demonstrated to control stem cell differentiation by providing controlled physical, chemical, and biological properties that can be utilized to regulate cell-matrix interactions[23,26,35,36]. To investigate whether our biodegradable $MnO_2$ hybrid nanoscaffolds have an enhanced binding affinity toward ECM proteins to promote cell adhesion, neuronal differentiation of stem cells, and neurite outgrowth through the ECM-mediated integrin signaling pathway, we first investigated the interaction between 2D $MnO_2$ nanosheets and laminin proteins (Fig. 2a-b, Supplementary Fig. 1). Using a bicinchoninic acid (BCA) assay, we observed significantly increased laminin adsorption on $MnO_2$ nanosheets (7.5-fold increase) compared to its binding toward control glass and polymer substrates (Fig. 2c). To better understand the origin of such strong binding interactions between ECM proteins and $MnO_2$-nanosheets, we used the density functional theory (DFT) method to calculate the binding energies between the $MnO_2$-nanosheets and a series of functional groups commonly exhibited in ECM proteins (Fig. 2b, d, e). The calculation results showed that electrostatic and polar-π interactions are the main contributors to the strong binding interactions of the biomolecules onto the $MnO_2$-nanosheets. For example, the binding energies for methylamine and methylbenzene are about 3-fold higher than that of water (Fig. 2e, Supplementary Fig. 2, Supplementary Table 1). Considering laminin proteins are rich in amino and aromatic functional groups, the DFT calculation results indicated that these interactions are critical for the strong binding of ECM proteins onto the $MnO_2$-nanosheet. Given the extraordinarily high crystal surface of 2D $MnO_2$ nanosheets, we speculated that the nanoscaffolds would also have strong binding interactions toward small molecule drugs that contain aromatic and amine structures. Our DFT calculation approach was thus further utilized to provide insight into the laminin-induced formation of 3D $MnO_2$ hybrid nanoscaffolds and acted as a screening method to identify neurogenic or anti-inflammatory drugs that can enhance survival and neuronal differentiation of NSCs in vitro and in vivo.

To study neuronal differentiation of stem cells using our $MnO_2$ hybrid nanoscaffolds, we synthesized layer-by-layer $MnO_2$ nanoscaffold assembly (3D-$MnO_2$ nanoscaffolds) using a vacuum filtration method that enabled us to generate highly homogeneous and reproducible 3D-$MnO_2$ nanoscaffolds (Fig. 2f). Compared to conventional 3D nanoscaffold-fabrication methods such as spraying, drop-casting, and electrochemical deposition, our applied vacuum filtration method can produce large-scale, homogeneous, free-standing, and mechanically robust 3D nanoscaffolds in a highly controllable way (Supplementary Figs. 3, 4). To perform the 3D-$MnO_2$ nanoscaffold-based stem cell assay, we chose human induced pluripotent stem cell (hiPSC)-derived NSCs as a model system since hiPSC-derived NSCs can be effectively translated into clinical applications for neurodegenerative diseases and injuries[37].

By seeding hiPSC-NSCs on laminin-coated 3D-$MnO_2$ nanoscaffolds, we observed a significant enhancement of neuronal differentiation (43% increase) and neurite outgrowth (11-fold increase) compared to the control conditions by measuring the biomarker protein and gene expression levels (Fig. 2g-i, Supplementary Figs. 5, 6). To understand the underlying mechanism of the 3D-$MnO_2$ nanoscaffold-based enhanced neuronal differentiation and neurite outgrowth, we investigated the relevant laminin-mediated focal adhesion-dependent signaling pathways using a qRT-PCR (quantitative reverse transcription-polymerase chain reaction, Supplementary Table 2) technique. Indeed, a substantial increase of focal adhesion kinase (FAK) gene (4.7-fold) and an upregulation of a neuronal growth

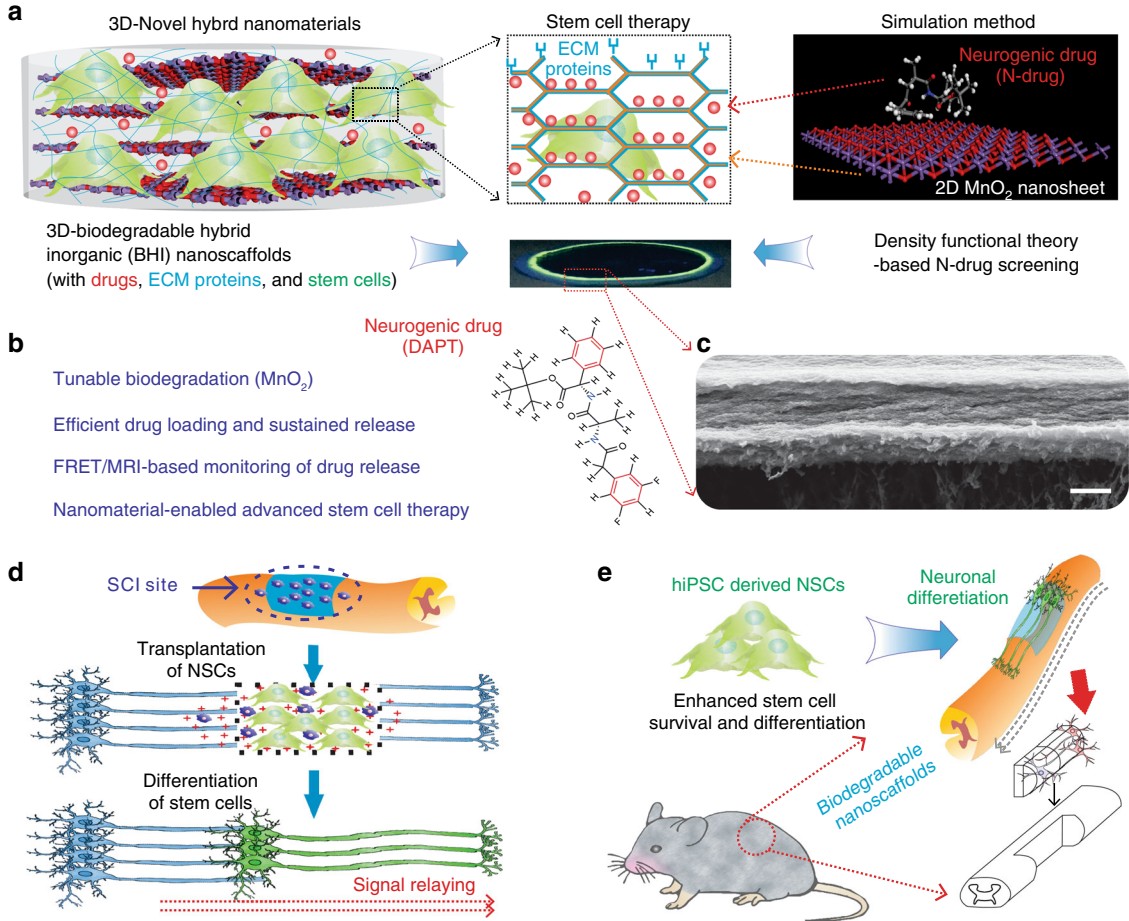

**Fig. 1** BHI nanoscaffolds for advanced stem cell therapy. **a** To develop an effective method for stem cell transplantation, we synthesized a BHI nanoscaffolds that simultaneously integrate advancements in 3D-hybrid nanomaterials and DFT calculations-based precision drug screening. Cells are labeled with green due to their green fluorescence protein labeling. Laminin proteins are colored in blue. Drugs are represented by red-colored dots. In the simulation scheme, blue colored atoms represent manganese and red colored atoms represent oxygen. **b** Compared to conventional inorganic scaffolds for stem cell transplantation, our BHI nanoscaffold self-assembled from atomic-thin $MnO_2$ nanosheets, ECM proteins, and therapeutic drugs has unique advantages including: (i) Redox mediated tunable biodegradation; (ii) Efficient drug loading and sustainable release; (iii) FRET/MRI monitorable drug release; (iv) Nanomaterials enabled advanced stem cell transplantation. **c** A representative SEM (Scanning Electron Microscopy) image of BHI nanoscaffolds. **d–e** The unique advantages from our innovative BHI nanoscaffold effectively improved the stem cell transplantation under CNS injured microenvironments, which typically have highly inflammatory and inhibitory microenvironments at the injury site. Blue colored and elongated cells indicate host neurons; dark-blue cells with pink nuclei: immune cells such as macrophage; red cross: inflammatory and inhibitory microenvironments. Specifically, the significantly enhancement of stem cell transplantation from BHI nanoscaffold is hypothesized to achieve through an improved cell adhesion and neuronal differentiation. A murine hemisection SCI model was used to evaluate the in vivo survival and differentiation of BHI nanoscaffold-transplanted iPSC-NSCs. Scale bar: 500 nm

cone-associated *GAP43* gene (36%) were observed from hiPSC-NSC-derived neurons on 3D $MnO_2$ nanoscaffolds, compared to those cultured on a glass substrate (Fig. 2g-i, Supplementary Figs. 6–8, Supplementary Methods). In short, these results strongly suggested that our 3D-$MnO_2$ nanoscaffolds can improve neuronal differentiation and neurite outgrowth, through the enhanced laminin binding and focal adhesion-related pathways.

**Controllable biodegradation of $MnO_2$ hybrid nanoscaffolds.** While low-dimensional inorganic nanomaterials have shown great potential in stem cell-based regenerative medicine, in vivo biocompatibility and biodegradation of these nanomaterials are the most critical issues to be addressed before inorganic nanomaterial-based stem cell applications can be fully realized. To demonstrate the tunable biodegradation study of $MnO_2$ nanoscaffolds in extracellular microenvironments, we first investigated the degradation of 2D-$MnO_2$ nanosheets using aqueous solution of ascorbic acid (vitamin C, Fig. 3a,

Supplementary Fig. 9). UV-Vis absorption spectrum data confirmed that 2D-$MnO_2$ nanosheets were degraded by ascorbic acid in a dose-dependent manner. Similarly, a controllable degradation rate of $MnO_2$ nanosheets by ascorbic acid was observed using micropatterned-$MnO_2$ nanoscaffolds, by directly monitoring the disappearance of the micropatterned-$MnO_2$ nanoscaffolds and by analyzing the x-ray energy dispersive spectroscopy (EDS) data (Fig. 3c, d, Supplementary Methods). In addition, the tunability of biodegradation rate can be effectively achieved by changing the number of assembled layers (Supplementary Fig. 10). Furthermore, we investigated the redox properties of $MnO_2$ nanoscaffolds in PBS using cyclic voltammetry (CV) to study the degradation without exogenous bioreductants. We could detect a clear reduction voltage peak at −700 mV from the CV curves, at which $MnO_2$ nanoscaffolds (Fig. 3b) degrade. From these electrochemical experiments, we confirmed our hypothesis that our synthesized $MnO_2$ nanoscaffolds could be degraded via an unconventional redox-mechanism. In parallel, we inserted $MnO_2$

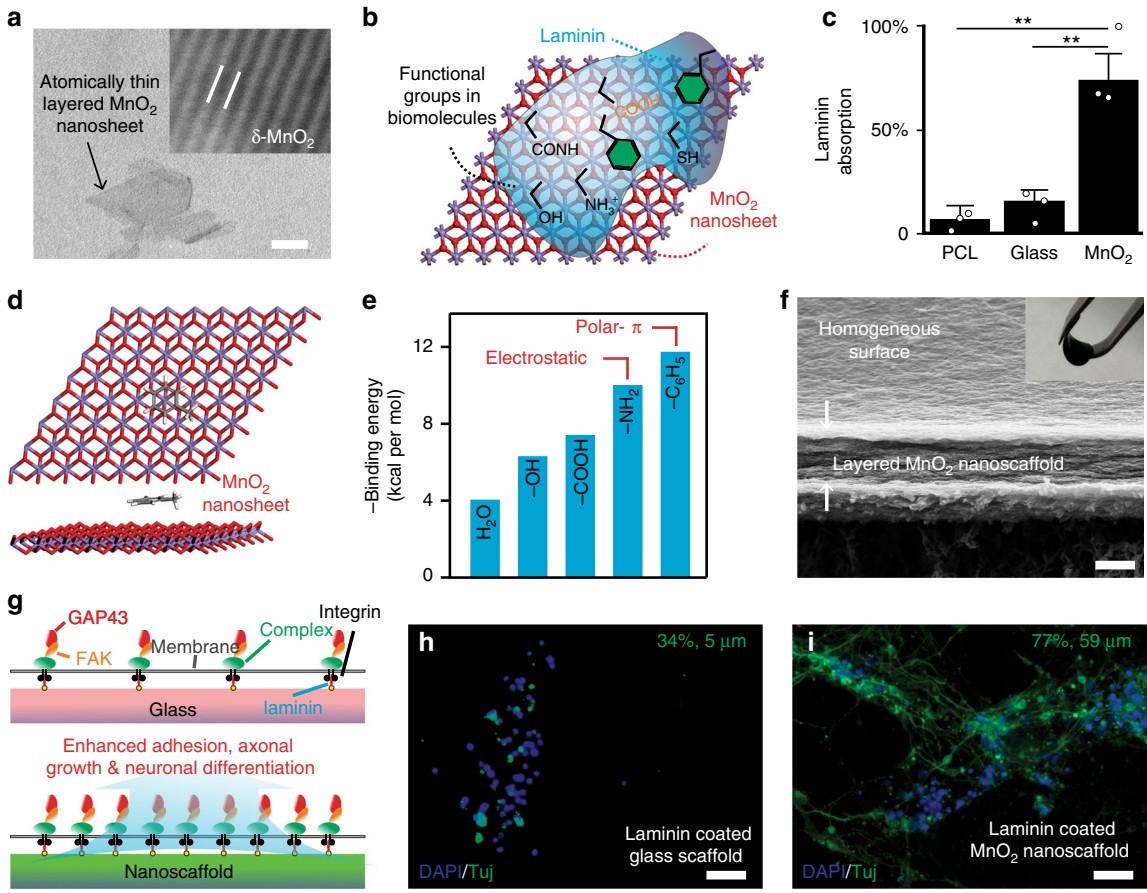

**Fig. 2** Enhanced stem cell differentiation using biodegradable $MnO_2$ hybrid nanoscaffolds. **a** Representative TEM (Transmission Electron Microscopy) of atomic-thin layered $MnO_2$ nanosheets. Inset: HR (High Resolution) TEM image (image size: 18 Å by 14 Å). **b** A schematic diagram describing the intermolecular binding between $MnO_2$ nanosheets and selected functional groups that are commonly existent in ECM proteins and biomolecules. **c** A significantly upregulated ECM protein (laminin) binding towards 2D $MnO_2$ nanosheet, compared to control substrate [etched glass and polycaprolactone (PCL) substrates). These upregulated laminin binding was studied by a BCA protein assay. Data are mean ± s.d. $n = 3$, **$P < 0.01$ by one-way ANOVA (Analysis of variance) with Tukey post-hoc test. **d**, **e**, By modeling small molecule-nanosheet interactions (**d**), we summarized differential binding affinities of $MnO_2$ nanosheets towards of a library of functional groups (**e**). These simulation results aligned well with experimentally observed enhanced laminin binding, as laminin structures are rich in both amino and aromatic moieties. Exemplary molecule in **d**: toluene. Carbon and hydrogen atoms are colored in black and gray, respectively. **f** A representative SEM image showing the layered structure and homogeneous surface of $MnO_2$ hybrid nanoscaffolds fabricated using the vacuum filtration technique. Inset photograph: a free-standing $MnO_2$ hybrid nanoscaffold fabricated by vacuum filtration and manipulated by a tweezer. **g** A schematic diagram illustrating the proposed mechanism for the enhanced neuronal differentiation on $MnO_2$ hybrid nanoscaffolds. Red color indicates GAP43; Orange indicates FAK; Green indicates integrin complex. **h**, **i** Representative immunostaining data supporting the significantly enhanced neuronal differentiation and neurite outgrowth of iPSC-NSCs on $MnO_2$ hybrid nanoscaffolds (**i**) compared to control substrates (**h**). Cell nuclei are in blue and TuJ1 is pseudocolored in green. Average percentages of neuron and neurite lengths were measured using NeuronJ software (**h**: $n = 9$; **i**: $n = 12$) and labeled on the top right of each image. Scale bars: **a** 25 nm; **f** 500 nm; **h**, **i** 100 μm

nanoscaffolds in between two layers of cells, which can mimic in vivo transplantation conditions, to study the nanoscaffold degradation profiles, as well as to investigate whether such redox-mediated biodegradation of $MnO_2$ nanoscaffolds was possible in tissue-mimicking conditions without adding any exogenous bioreductants or electrical stimuli. As a negative control experiment, graphene oxide (GO) nanoscaffolds were also inserted in between two layers of cells. The biodegradation of both nanoscaffolds ($MnO_2$ vs. GO) was examined daily by measuring the thickness of the dark-colored nanoscaffold layers. Consistent with previous reports, we did not observe any noticeable degradation of GO nanoscaffolds throughout a month-long observation (Fig. 3e)[38]. In contrast, $MnO_2$ nanoscaffolds rapidly degraded with over 30% of the $MnO_2$ nanoscaffolds degraded within one week, and a half-degradation time was around 2 weeks (Supplementary Fig. 9). This result proved that the biodegradability of $MnO_2$ nanoscaffolds could be induced by cells without

delivery of exogenous reductants. Moreover, we could control the degradation rate of $MnO_2$ nanoscaffolds by showing a tunable half-degradation period from a few minutes to one month. This tunability of a biodegradation rate was achieved by changing the assembled layered-structures of $MnO_2$ nanosheets and by controlling concentrations of reductants (Fig. 3f, Supplementary Fig. 10). In short, our developed $MnO_2$ nanoscaffolds represent an innovative inorganic hybrid nanoscaffold system that can be biodegraded in vitro. Given that CNS microenvironment contains highly concentrated bioreductants and different degradation speeds would be needed for different applications, the controllable biodegradation properties of $MnO_2$ nanoscaffolds could be more appealing and important in the field of neural tissue engineering[39].

**3D $MnO_2$ hybrid nanoscaffolds self-assembled with ECM protein**. One of the critical issues of conventional degradable

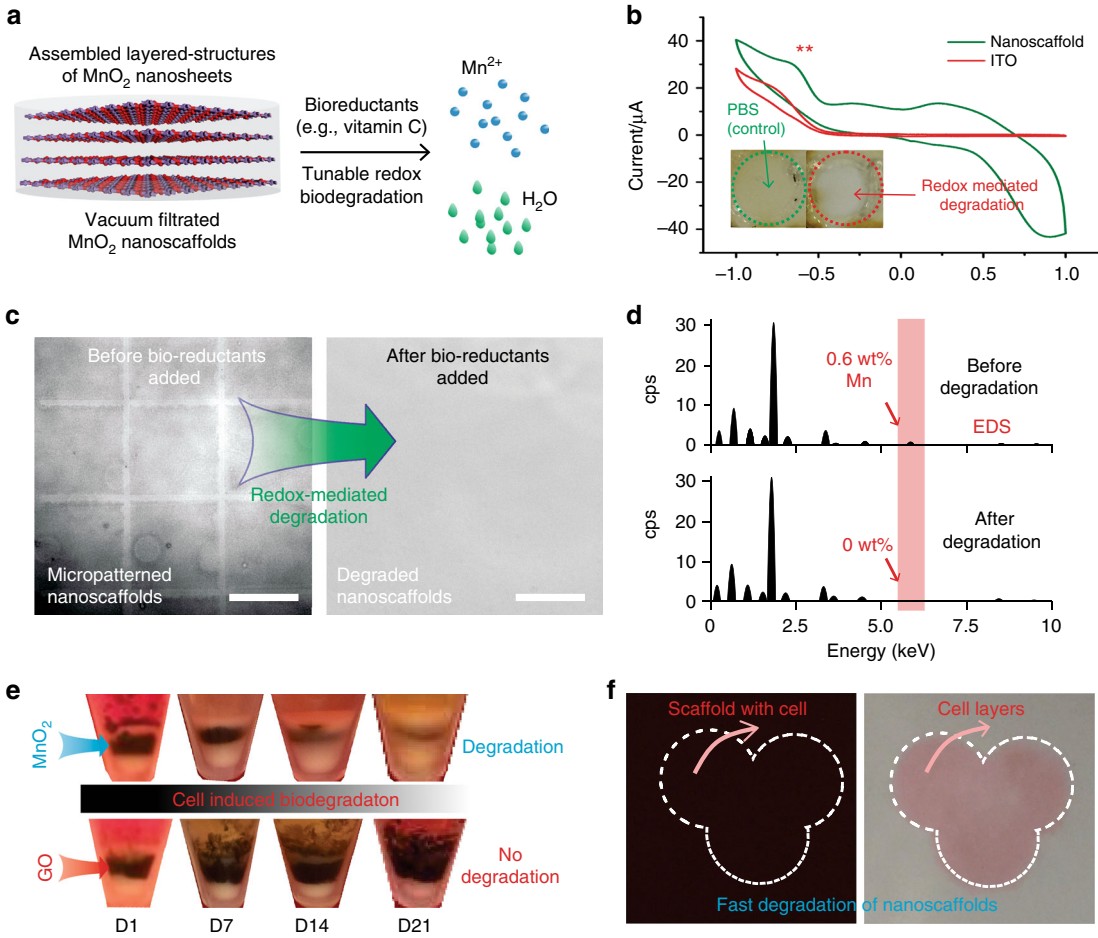

**Fig. 3** Controllable biodegradation of MnO$_2$ hybrid nanoscaffolds. **a** A schematic diagram explaining an unconventional redox biodegradation mechanism of our MnO$_2$ hybrid nanoscaffolds. This redox biodegradation can be achieved through either bioreductants exists commonly in human body such as vitamin C, electrically delivered reduction signals or by stem cells cultured with the nanoscaffold. Degradation products include water (colored in green) and Mn$^{2+}$ (colored in blue). **b** Controllable redox biodegradation of MnO$_2$ hybrid nanoscaffolds demonstrated through cyclic voltammetry. A successful degradation of nanoscaffolds were confirmed by the disappearance of yellow color from nanoscaffold triggered by electrical stimuli. x axis indicates voltage (v). **c**, **d** Degradation of nanoscaffold by commonly existent bioreductants (e.g., vitamin C), indicated by the decay of micropatterns from the micropatterned nanoscaffold (**c**); and the disappearance of manganese elements from the substrate after degradation through EDS analysis (**d**). cps means count per scan. **e** Time dependent biodegradation of MnO$_2$ hybrid nanoscaffolds in cell culture without addition of any external trigger. The degradation of nanoscaffold was examined based on the thickness of the black-colored layer sandwiched between two layers of cells. GO nanoscaffolds were used as a negative control and no noticeable degradation was observed. Half-degradation time of MnO$_2$ hybrid nanoscaffolds was determined to be around 2 weeks. **f** Controllable fast-biodegradation of MnO$_2$ hybrid nanoscaffolds. By controlling bioreductant (vitamin C) concentrations, a fast degradation of iPSC-NSC seeded nanoscaffold and formation of iPSC-NSC sheet was achieved. Nanoscaffold is indicated by the dark-colored background before degradation. Size of each image is 1 cm by 1 cm. Cells were stained with pink color for convenience of observation. Scale bar: **c** 100 μm

bioscaffolds is degradation-mediated disruption of cellular microenvironments, which can interrupt continuous neuronal differentiation and neurite outgrowth of transplanted NSCs[15,40]. To this end, biocompatible 3D bioscaffolds complexed with ECM proteins or peptides, such as laminin, fibronectin, and Arginylglycylaspartic acid (RGD), that enhance neuronal differentiation of stem cells and neurite outgrowth continuously, have provided a promising solution for advanced stem cell-based tissue engineering[35,41,42]. As such, inspired by a recent report on the non-covalent preparation of hydrogels[43], we developed a method to generate biocompatible 3D-MnO$_2$ hybrid nanoscaffolds complexed with laminin-protein, termed, 3D-MnO$_2$-laminin hybrid nanoscaffolds. Interestingly, MnO$_2$-laminin hybrid nanoscaffold, an innovative 3D-inorganic scaffold self-assembled from laminin, was successfully synthesized by mixing 2D-MnO$_2$ nanosheets with laminin solutions (Fig. 4a–e, Supplementary Fig. 11). The self-assembly process could be achieved by the strong interactions

of laminin toward 2D-MnO$_2$ nanosheets, where laminin can function as adhesive layers (binder) for individual MnO$_2$ nanosheets. To investigate whether 3D-MnO$_2$-laminin hybrid nanoscaffolds promote the neuronal differentiation of NSCs and the following neuronal behaviors including neurite outgrowth, we performed stem cell assays using three different substrates/scaffolds (glass, MnO$_2$ nanoscaffolds, and 3D-MnO$_2$-laminin hybrid nanoscaffolds) under the same culture conditions. After 6 days of stem cell differentiation assays, we found dramatically higher cell counts from our 3D-MnO$_2$-laminin hybrid nanoscaffolds compared to glass (740% higher) and MnO$_2$ nanoscaffolds (270% higher) controls (Fig. 4b, f–h, Supplementary Fig. 11). Furthermore, through a neuronal marker, beta-III tubulin (TuJ1) immunostaining, we confirmed even more significant improvement of neuronal differentiation and neurite outgrowth from our 3D-MnO$_2$-laminin hybrid nanoscaffolds compared to the other control substrates, showing 11 times longer average neurite

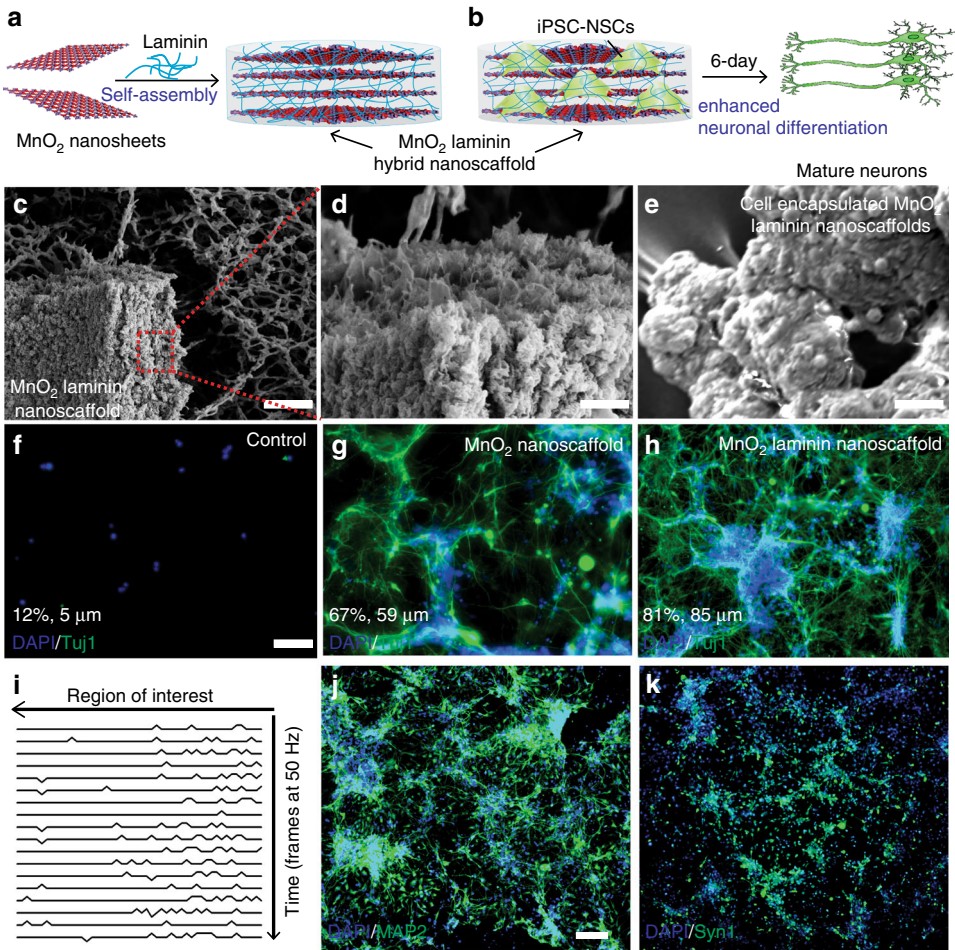

**Fig. 4** 3D $MnO_2$ hybrid nanoscaffolds self-assembled with ECM proteins. **a**, **b** Schematic diagram illustrating the mechanism for the self-assembly of ECM protein (laminin) and $MnO_2$ nanosheets through a non-covalent crosslinking mechanism. This mechanism was utilized to synthesize $MnO_2$ laminin hybrid nanoscaffolds, and iPSC-NSCs cultured on $MnO_2$-laminin hybrid nanoscaffolds successfully differentiated into mature (MAP2+) neurons after 6 days (**b**). **c–e** Representative SEM images of $MnO_2$ laminin hybrid nanoscaffolds (**c**, **d**), and iPSC-NSC-encapsulated hybrid nanoscaffolds (**e**). **f–h** Immunostaining results on neuronal markers (TuJ1, labeled with green) demonstrate significant enhancements of cell adhesion and neuronal differentiation of iPSC-NSCs differentiated on $MnO_2$ laminin hybrid nanoscaffolds (**h**) compared to both control (glass, **f**) substrates and $MnO_2$ hybrid nanoscaffolds (**g**). Average percentage of neuron and neurite lengths were labeled on the bottom left of each image. **i** Time-lapse calcium imaging results of iPSC-NSC differentiated neurons on $MnO_2$ laminin hybrid nanoscaffolds. **j**, **k** representative immunostaining images on mature neuronal markers (**i**: MAP2 and **k**: Synapsin 1, labeled with green) from iPSC-NSC-differentiated neurons on $MnO_2$ laminin hybrid nanoscaffolds. Scale bars: **c** 10 μm; **d** 500 nm; **e–h** 100 μm; **j**, **k** 100 μm

lengths compared to laminin-coated glass and 1.4 times longer than laminin-coated $MnO_2$ nanoscaffold (Fig. 4f, Supplementary Fig. 6). Meanwhile, we have also verified the neurons formed on our hybrid nanoscaffolds are mature through time-dependent calcium imaging technique and immunostaining with mature neuronal markers such as Microtubule-associated protein 2 (MAP2) and Synapsin (Syn1) (Fig. 4i-k, Supplementary Fig. 7). These results support our hypothesis that 3D-$MnO_2$-laminin hybrid nanoscaffolds can effectively and steadily promote neuronal differentiation of stem cells and neuronal behaviors for versatile stem cell therapies.

**Spatiotemporal controlled delivery and monitoring of drugs**. While some conventional biodegradable and biocompatible 3D-hybrid scaffolds have shown their potential to promote stem cell neuronal differentiation and neurite outgrowth, there is still a lot of room for improvement to control stem cell differentiation and neuronal behaviors in a more selective and temporally controlled manner in vivo[44–47]. These requirements would be essential to achieve the full therapeutic potential of transplanted stem cells for SCI treatment. Addressing this challenge, spatiotemporal

controlled delivery of soluble cues such as small organic molecules (e.g., neurogenic drugs to selectively induce stem cell neuronal differentiation) using our 3D-hybrid inorganic nanoscaffolds provides a promising solution[44,46,47]. Conventional scaffolds that typically use physical encapsulation to load drugs normally suffer from rapid diffusion of drugs, which leads to undesired damage to the transplanted cells, as well as the surrounding tissues because of the high drug concentration initially, and limited neurogenic effect later on due to an insufficient remaining drug concentration[45–47]. To this end, our developed 3D-$MnO_2$-laminin hybrid nanoscaffolds showed improved drug-loading capability and minimized burst-release owing to strong drug-binding affinity to the nanoscaffolds. For a comprehensive study of drug loading and monitoring of drug release using our nanoscaffolds, we first used a fluorescent aromatic ring-containing small molecule, Rhodamine B (RhB), as a model drug system. To optimize the loading and binding of drug molecules, RhB was first loaded onto 2D-$MnO_2$-nanosheets. Then, the RhB-loaded $MnO_2$-nanosheets self-assembled with laminin to generate 3D-$MnO_2$-laminin hybrid nanoscaffolds (Fig. 5a). We adapted a quantitative fluorescence resonance

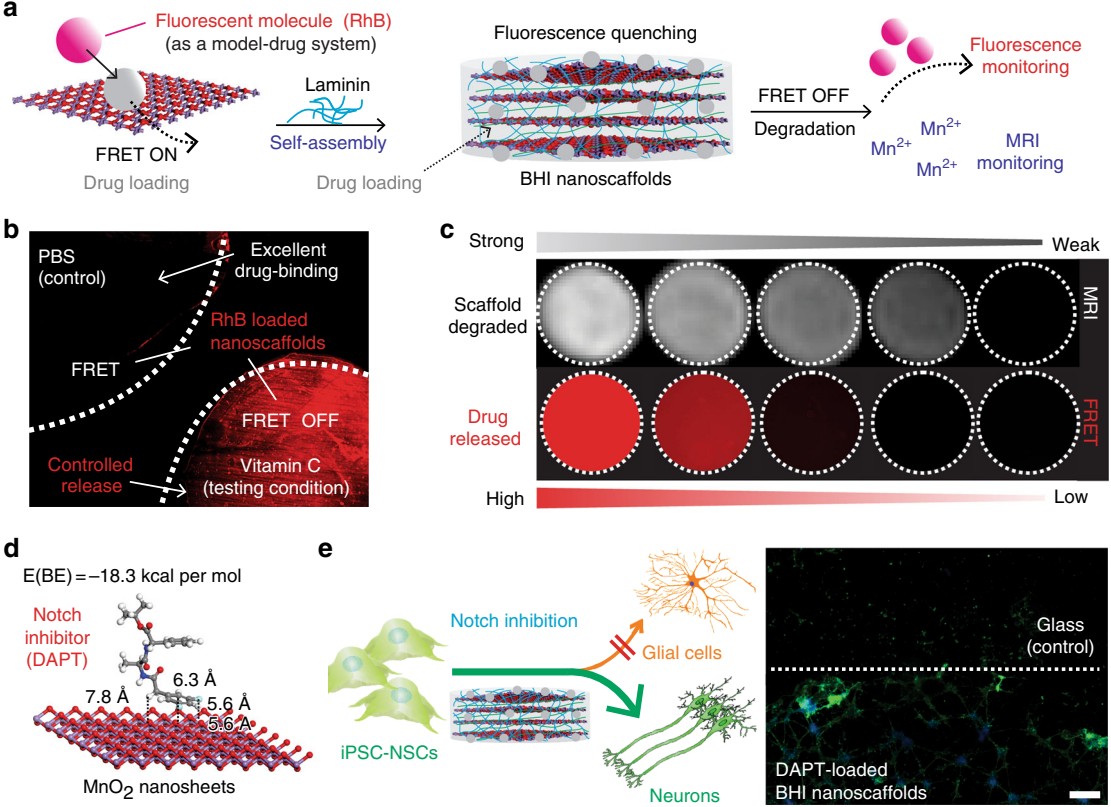

**Fig. 5** Spatiotemporal controlled delivery of soluble cues using 3D-hybrid inorganic nanoscaffolds. **a** Schematic diagram of drug loading, releasing, and monitoring on the MnO$_2$ laminin hybrid nanoscaffolds. Drug (pink and gray circles) was first loaded onto individual nanosheets, then self-assembled with laminin to form the drug loaded-MnO$_2$ laminin hybrid nanoscaffolds to achieve controlled drug release. Two modalities of monitoring scaffold degradation and drug release can be achieved by the nanoscaffold through FRET and the stoichiometrical release of T1 active Mn$^{2+}$, respectively. **b** Fluorescence microscopy images demonstrating excellent drug hold-up from drug-loaded hybrid nanoscaffolds, and controlled drug release by bioreductants. Release of model drugs were monitored by the red fluorescence nearby nanoscaffold. **c** A MRI-based monitoring of drug release enabled by our hybrid nanoscaffold, which is confirmed by a direct correlation between the amount of released drug (indicated by red fluorescence) and T1 MRI intensities detected from the same nanoscaffolds. This unique drug monitoring was enabled by the strong interaction between drug and nanoscaffold, which determines the stoichiometrical relevance between amount of drug and Mn$^{2+}$ released as degradation products. 5, 2.5, 1, 0.5, 0.1 mg (left to right) of scaffold were degraded before MRI and fluorescence measurements. Samples were incubated for 2 days before imaged. Dotted circles have diameters of 1 mm. **d** An optimized binding geometry and binding energy of simulation-screened neurogenic drug (DAPT) toward nanosheets. Nitrogen atoms are colored in blue. **e** Spatial control of neuronal differentiation and neurite outgrowth across the boundary between control substrates and DAPT-loaded MnO$_2$ laminin hybrid nanoscaffolds. DAPT-loaded MnO$_2$ laminin hybrid nanoscaffolds enhanced neuronal differentiation of iPSC-NSCs compared to control substrate and MnO$_2$ hybrid nanoscaffolds. Through a spatial patterning of DAPT-loaded MnO$_2$ laminin nanoscaffolds, a spatially controlled neuronal differentiation was successfully demonstrated. TuJ1 and nuclei staining of iPSC-NSC were indicated by green and blue, respectively. Astrocytes are colored in orange in the scheme. Scale bar: **e** 100 μm

energy transfer (FRET)-based approach to monitor released or non-binding RhB molecules[32]. Our FRET-based method allowed us to assay the drug loading and release process (Fig. 5a). Based on this FRET-based drug monitoring method, we observed an excellent drug-binding affinity onto the 3D-MnO$_2$-laminin hybrid nanoscaffolds, wherein minimal RhB release from the nanoscaffolds was detected over 7 days. However, as soon as we introduced bioreductant (vitamin C) to the RhB-loaded 3D-MnO$_2$-laminin hybrid nanoscaffolds, the fluorescence signal of RhB release was observed with an over 500-fold increase with a sustainable delivery profile (Fig. 5b, Supplementary Figs. 12–13). This experimental result strongly supports that our 3D-hybrid inorganic nanoscaffold-based drug delivery platform can control drug release kinetics over a few weeks through degradation of nanoscaffolds. Additionally, the stoichiometrically equivalent Mn$^{2+}$ ion release and the MnO$_2$ degradation (1:1 ratio) encouraged us to hypothesize that MRI signals from Mn$^{2+}$ can be utilized to quantify the degradation rate of our hybrid nanoscaffolds and to correlate the intensity of MRI signal with the

amount of drug released (Fig. 5a, c). Indeed, by inducing the nanoscaffold degradation by bioreductants, we found that the amount of released drug, measured by the fluorescence intensity of RhB was closely correlated with the intensity of MRI signal (Fig. 5c). This "on/off" MRI-based monitoring of drug release has not yet been demonstrated in conventional scaffolds, thereby offering a tool that can provide a much-improved investigation on drug delivery and in vivo release[48].

The optimized condition regarding drug loading and release, based on the fluorescent RhB molecule as a model drug, was used to load and deliver neurogenic drugs for an enhanced neuronal differentiation. To screen the optimal neurogenic drug, we applied the DFT calculations (Fig. 2d, Supplementary Table 1) and selected a neurogenic drug (DAPT: N-[N-(3,5-Difluorophenacetyl)-L-alanyl]-S-phenylglycine t-butyl ester) based upon its high binding energy to 2D-MnO$_2$ nanosheet (Fig. 5d, Supplementary Figs. 13–14). DAPT is a γ-secretase and Notch inhibitor that simultaneously promotes neuronal differentiation and neurite outgrowth while suppressing astrocyte differentiation[49,50].

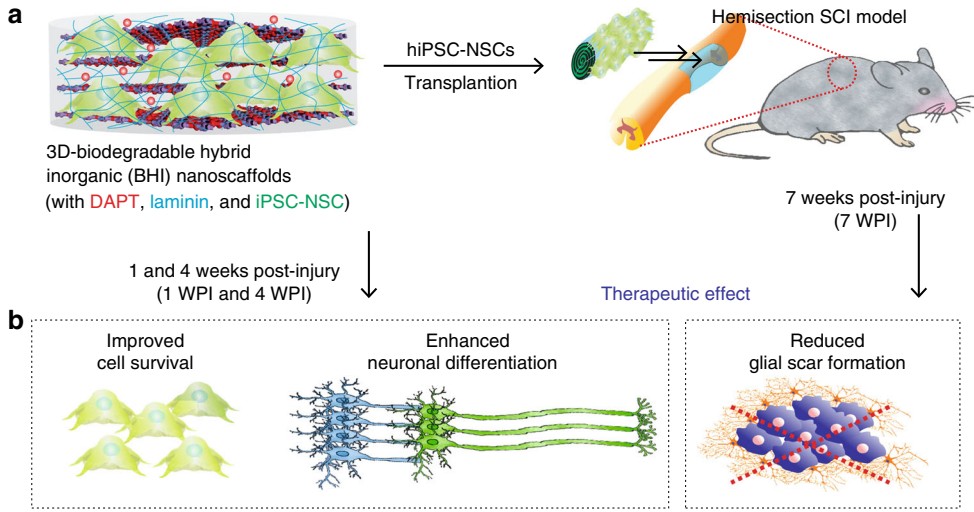

**Fig. 6** Pathways of enhanced stem cell transplantation into SCI sites by 3D-BHI nanoscaffolds. **a**, **b**, Schematic diagram (**a**) showing the enhanced transplantation of iPSC-NSC in a murine lateral hemisection SCI model by 3D BHI nanoscaffold and the proposed mechanisms (**b**) for the enhanced transplantation and potential beneficial effects on overcoming inhibitory microenvironments in the SCI sites

The calculated binding energy between DAPT and the 2D-$MnO_2$ nanosheet is $-18.3$ kcal mol$^{-1}$, an over 4-fold increase compared to the binding of solvent (water) to nanoscaffolds, indicating that DAPT drugs can be strongly adsorbed to the $MnO_2$ surface. Indeed, by forming DAPT-loaded 3D-$MnO_2$-laminin hybrid nanoscaffolds, the spectrums from matrix-assisted laser desorption/ionization (MALDI) time-of-flight (TOF) mass spectrometer showed a high amount of DAPT loaded onto the nanoscaffolds, while control scaffolds (glass and polymer substrate) did not show any noticeable peaks (Supplementary Fig. 14). To investigate the effect of DAPT-loaded 3D-$MnO_2$-laminin hybrid nanoscaffolds on stem cell neuronal differentiation and neuronal behaviors, we tested hiPSC-NSC-based neuronal differentiation assays using DAPT-loaded 3D-$MnO_2$-laminin nanoscaffolds and related controlled conditions for one week. We found a strong enhancement of neuronal differentiation from the DAPT-loaded 3D-$MnO_2$-laminin nanoscaffold condition (a 1.4-fold enhancement of *Tuj1* mRNAs and a 1.7-fold enhancement of neurite outgrowth compared to 3D-$MnO_2$ laminin hybrid nanoscaffolds), as well as suppressed *GFAP* (Glial fibrillary acidic protein) mRNA expression (Fig. 5e, Supplementary Figs. 15, 16). Remarkably, in the boundary of DAPT-loaded 3D-$MnO_2$-laminin hybrid nanoscaffold and glass, NSC-derived neurons across the boundary, cultured under the same condition, had a dramatic change in morphology and neurite outgrowth (Fig. 5e). This result provided a direct comparison between our 3D-$MnO_2$-laminin hybrid nanoscaffolds and conventional scaffolds and indicated the ability of spatiotemporal control of hiPSC-NSC differentiation using our drug-loaded 3D-hybrid inorganic nanoscaffolds.

**Enhanced stem cell transplantation into SCI sites**. With the prominent effects of the 3D-hybrid inorganic nanoscaffolds on improving the adhesion, neuronal differentiation of hiPSC-NSCs, and neurite outgrowth of differentiated neurons, we then further tested the effects of the nanoscaffold on enhanced stem cell transplantation in vivo (Fig. 6). To transplant the stem cell-seeded nanoscaffolds, we first generated a T10 thoracic hemisection lesion to the spinal cord of an adult mouse, then the hiPSC-NSC seeded-nanoscaffolds (as an experimental condition) and -polycaprolactone (PCL) polymer scaffolds (PCL-cell group, as a control condition) were rolled up and inserted into the

hemisected SCI lesion (Fig. 6a)[51]. To identify our transplanted cells, hiPSC-NSCs were genetically labeled with green fluorescent protein (GFP) (i.e., hiPSC-NSC-GFP). Surgifoam inserted mice with injuries were used as a control condition. After transplantation, we first evaluated nanoscaffold biodegradation in vivo by detecting the amount of degraded Mn (manganese) element in mouse urine samples using inductively coupled plasma mass spectrometry (ICP-MS) analysis (Supplementary Fig. 17). Consistent with our previous in vitro studies, we could observe rapid in vivo degradation of nanoscaffold. This degradation of transplanted nanoscaffolds was also detected by the color change (from black to brown) in a time-dependent manner throughout the first-week post-transplantation. After investigating the in vivo biodegradability, we then studied the effects of 3D-BHI nanoscaffold for enhanced stem cell transplantation. We hypothesized that our 3D-BHI nanoscaffold could robustly improve the survival of hiPSC-NSCs and the differentiation of hiPSC-NSCs into neurons in the early and intermediate stages (1–4 weeks post injury (WPI)) (Fig. 6, Supplementary Figs. 18–25). We also expected that the 3D-BHI nanoscaffold could reduce astroglial scar formation in the longer term (7 WPI) (Fig. 6). To test our hypothesis, we first performed short-term (1-week) in vivo stem cell transplantation assay in the injured site and 3 different control conditions for stem cell transplantation were included as controls (Fig. 7, Supplementary Figs. 20–23). From our immunostaining data, we observed significantly higher populations of GFP + cells and TuJ1 + cells from our testing condition compared to the other 3 control conditions, which directly suggests an enhanced stem cell transplantation. This result is also consistent with our Caspase 3 immunostaining data (Fig. 7, Supplementary Fig. 23), where fewer hiPSC-NSC-GFP cells transplanted by 3D-BHI nanoscaffold showed apoptotic markers as compared to the polymer scaffold-transplanted cells. Additionally, hiPSC-NSC-GFP cells transplanted by 3D-BHI nanoscaffold showed more spreading, whereas GFP + cells from two control conditions (Fig. 7) showed less spreading. This could be due to the promoted cellular adhesion and could improve neuronal differentiation, based on our in vitro focal adhesion studies (Supplementary Fig. 6). This trend of enhanced cell survival and increased neuronal differentiation of stem cells from 3D-BHI nanoscaffold-treated condition continued until 4-WPI, as shown in (Supplementary Fig. 24). Furthermore, the percentage of hiPSC-NSC-GFP with more mature neuronal markers (Syn1) is

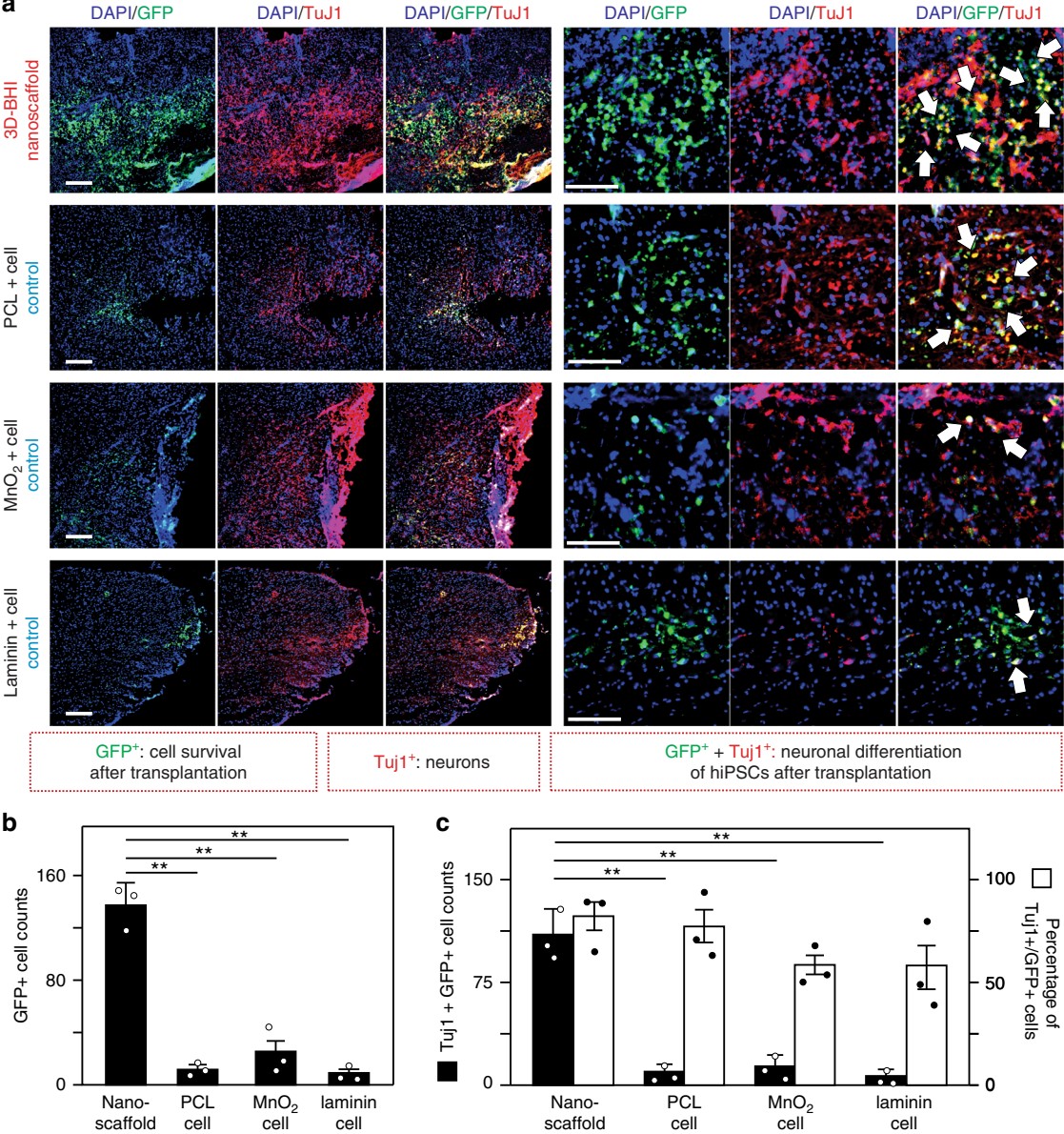

**Fig. 7** 3D-BHI nanoscaffold enhances survival and neuronal differentiation of hiPSC-NSC-GFP. **a** Immunohistological staining analysis was performed on tissue slices from 4 different animal groups transplanted with hiPSC-NSC-GFP to determine enhanced cell survival and improved neurogenesis from our 3D-BHI nanoscaffold transplanted condition. All tissue analysis was performed 1-week post transplantation and stained with DAPI (blue) and TuJ1 antibodies (red). Arrows indicate neuronal cells differentiated from hiPSC-NSC-GFP (identified by TuJ1+/GFP+ cells). **b** By quantifying the number of remaining GFP+ cells, the ability of 3D-BHI nanoscaffold to retain the significant higher amount of cells after transplantation was demonstrated by comparing to other cell transplantation groups. **c** Improved cell transplantation by our nanoscaffold can be further evidenced by an increased neuronal cell population and a higher percentage of neuronal cells in GFP+ cells (area = 1 mm$^2$). This is consistent with Supplementary Fig. 20, where co-labeling of GFP +/TuJ1+ and GFP+/GFAP+ nearby the injured sites suggest most of the transplanted hiPSC-NSC-GFP become neuronal cells but not astroglial cells. Scale bars: 100 μm. Error bars represent mean ± s.d.; $n = 3$, **$P < 0.01$ by one-way ANOVA with Tukey post-hoc test

also higher in 3D-BHI nanoscaffold-treated condition as compared to cell injection condition at this time point, indicating enhanced neuronal differentiation from our nanoscaffold-treated conditions (Supplementary Fig. 24). Equally importantly, the improved transplantation from our 3D-BHI treated conditions can mitigate inhibitory microenvironmental effects in the long-term (7-WPI) (Fig. 8). For example, immunostaining on a proliferation marker (PH3), inflammation marker (F4/80) and an astroglial scar marker (GFAP) revealed higher proliferation, suppressed inflammation and reduced astroglial scar formation from our 3D-BHI nanoscaffold, which could be largely attributed to the neurotrophic factors secreted by hiPSC-NSC-GFP and

DAPT released from nanoscaffold in the early stages[12]. Overcoming the highly inhibitory microenvironment such as inflammation and astroglial scars has been considered as an effective strategy to treat SCI[11]; and our results showed that 3D-BHI nanoscaffold could be utilized for the enhanced stem cell transplantation and for the treatment of CNS injuries. On the other hand, as future plans to apply our nanoscaffold for disease treatment, it would be essential to analyze long-term cell fates of transplanted stem cells and is critical to evaluate animal behavioral recovery from 3D-BHI nanoscaffold treated conditions using larger animal sets[52,53]. Nevertheless, given its unique properties demonstrated in vitro, and the enhanced

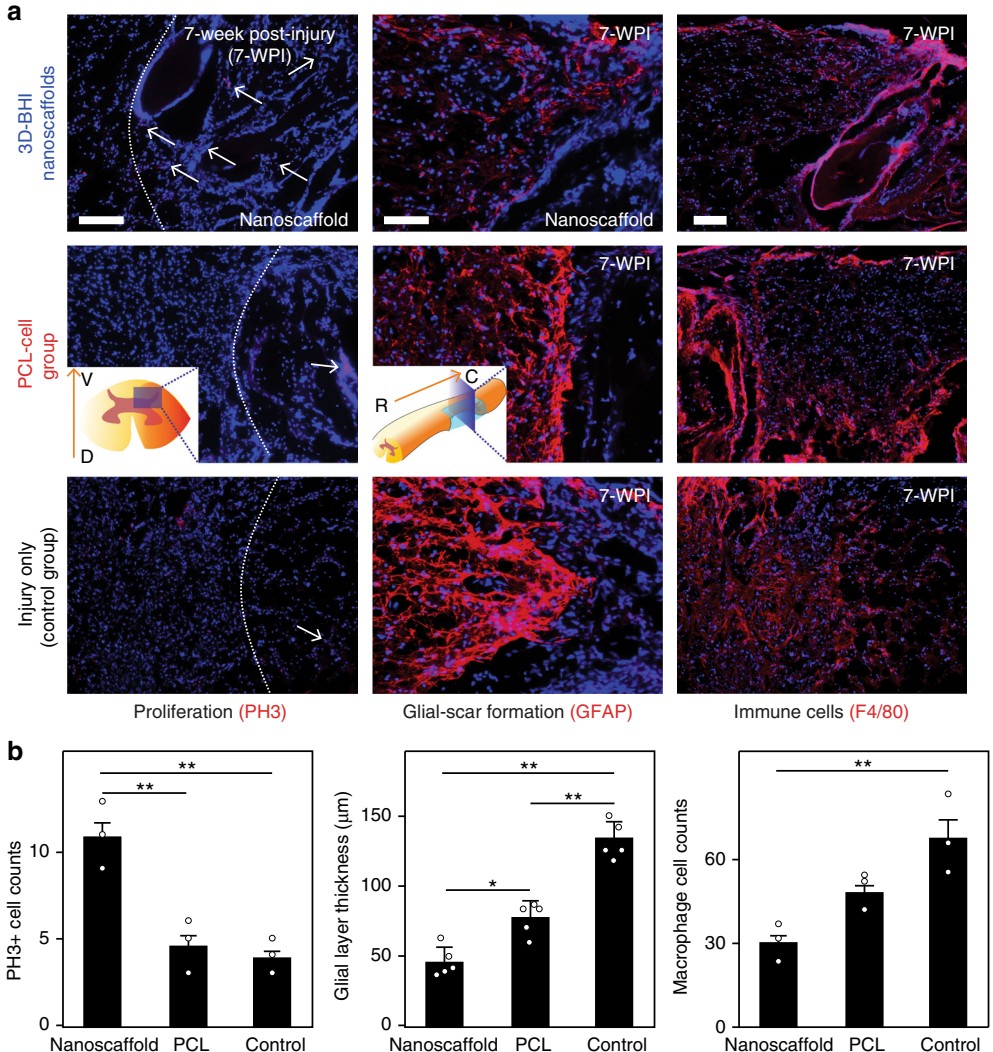

**Fig. 8** Improved long-term effects on SCI sites from 3D-BHI enhanced stem cell transplantation. **a**, **b** Histological immunostainings images (**a**) and quantifications (**b**) of cell proliferation markers (PH3), astroglial markers (GFAP), and immune cell markers (F4/80). These results collectively suggesting an improved stem cell transplantation from our 3D BHI nanoscaffolds can enhance cell proliferation inside scaffolds, reduce glial-scar formation and decrease infiltration of inflammatory cells in vivo at 7-WPI. Cell nuclei were colored in blue in **a**; other staining (PH3, GFAP, and F4/80) were in red. Scale bars in **a**: left and middle column: 200 μm; right column: 100 μm. Error bars represent mean ± s.d.; $n = 3$ or 5, $*P < 0.05$, $**P < 0.01$ by one-way ANOVA with Tukey post-hoc test

transplantation of hiPSC-NSCs in vivo compared to conventional cell transplantation methods, our developed 3D-BHI nanoscaffold could represent a promising candidate for stem cell therapy and for advanced stem cell transplantation.

## Discussion

The development and the use of biomaterials for stem cell-based tissue engineering to treat CNS diseases/injuries to date have focused on: (i) providing favorable microenvironments for endogenous and exogenous cellular regeneration and (ii) serving as a spatiotemporally controlled drug release platform to regulate pro-neuroregenerative signaling pathways. This work is based on the development of a biodegradable hybrid inorganic nanoscaffold and its utilization for the enhanced transplantation of stem cells into SCI sites. Our demonstrated nanoscaffold technology platform can be combined with other neurogenic drugs, as well as stem cell therapeutic efforts currently in development. In the developed hybrid nanoscaffold, we can tune the 2D/3D structural

properties of porous scaffolds and the unique physiochemical properties of $MnO_2$ nanoscaffold such as magnetic properties and degradation rate in a single platform. Recently, 2D/3D inorganic and carbon nanomaterials have attracted much attention, as they have the great potential to control stem cell neuronal differentiation and neuronal behaviors. However, several pertinent barriers including limited biodegradability and drug loading/release capability hinder their broad application toward stem cell-based tissue engineering. Thus, our developed biodegradable hybrid inorganic nanoscaffold-based stem cell therapeutic approach would be a useful tool for improving stem cell survival and inducing neuronal differentiation in vivo, and thus can provide insights into stem cell behaviors post-transplantation that may lead to novel therapies for the treatment of the neurodegenerative diseases. Collectively, our developed hybrid nanoscaffold-based approach to control stem cell differentiation into neurons and promote neurite outgrowth would provide an alternative to help overcome the critical barriers that limit cellular therapies for many devastating injuries and diseases.

## Methods

**Synthesis and characterization of MnO₂ nanosheet and GO.** MnO₂ nanosheets were synthesized based on a previous protocol with minor modifications[54]. Briefly, 2.2 g Tetramethyl ammonium pentahydrate (TMAOH·5H₂O, Alfa Aesar) was first dissolved in 20 ml 3% wt. H₂O₂ (Sigma-Aldrich) by vortexing (concentration of TMAOH is 0.6 M). In parallel, 0.594 g MnCl₂·4H₂O (Sigma-Aldrich) was dissolved in 10 ml de-ionized water (Concentration of MnCl₂ is 0.3 M) through sonication. Then the TMAOH dissolved in H₂O₂ solution was rapidly added into MnCl₂ solution within 10 s with fast stirring at 1200 r.p.m. (round per minute). Please note that gas will be generated and rapid increase of solution volume will be observed. The solution was continued to be stirred at 600 r.p.m. overnight and centrifuged at 2000×g for 5 min to obtain the bulk δ-MnO₂. After washing with water for 3 times and ethanol for 2 times through shaking and centrifuge, bulk MnO₂ was dried in the oven under ambient conditions for 12 h. After adding 100 mg into 10 ml de-ionized water, the solution was extensively sonicated in the Sonics bath sonicator for 10 h. Lastly, the solutions were centrifuged at 8801×g for 10 min to get rid of the aggregations and un-exfoliated products. The black solution was measured with concentration by evaporating water in the solution. MnO₂ nanosheet was diluted to 10 μg per ml for TEM (80 kv on a Philips CM12 with an AMT digital camera model XR111) and Ultra-Stem imaging. For X-ray photoluminescence spectroscopy (Thermo Scientific ESCALAB 250 Xi with a base pressure <1×10⁻⁹), MnO₂ nanosheet solution (100 μg per ml) was drop-casted onto a silicon substrate and dried in the vacuum. An Al-Kα monochromated X-ray source was used to obtain the core level spectra, and the instrumental broadening was around 0.5 eV. The hydrodynamic size and zeta potential of MnO₂ nanosheets in aqueous solution were measured by a ZS (Nano Zetasizer) dynamic light scattering instrument (Malvern Instruments, Malvern, UK), with the temperature set at 25 °C and a detection angle of 90 degrees. UV-Vis absorption spectrum of MnO₂ nanosheet solution was measured by a Varian Cary 50 spectrophotometer using a quartz cuvette.

Graphene oxide was synthesized based on our previous publications[55,56]. 1.0 gram of Graphite (Bay Carbon) was preoxidized in the mixture of sulfuric acid (Sigma-Aldrich, 98%), phosphor oxide (Sigma-Aldrich) and potassium persulfate (Sigma-Aldrich) at 80 degrees for overnight. Then the pre-oxidized graphite was washed with water, dried and reacted with sulfuric acid and potassium permanganate through a 3-step process. After quenching with H₂O₂, a shining gold solution appear, and the graphite oxide was purified with 10% HCl solution (Sigma-Aldrich) and water extensively. Lastly, graphite oxide was exfoliated into graphene oxide by tip sonication (Branson). Multi-layered graphene oxide was centrifuged down at 16,639×g for 45 min, and the final suspension of single or few layered graphene oxide was obtained.

**Measurement of protein absorption by MnO₂ nanosheet.** Into the solutions of ECM protein (laminin protein from Sigma-Aldrich, stock concentration of 200 μg per ml, 0.5 ml, PBS is from Thermo Fisher), 10 μl of MnO₂ nanosheet aqueous solution (3 mg per ml) was added, or a piece of etched glass (first treated by pirahana solution for 1 h, then oxygen plasma treated for 1 min, followed by polylysine (1 mg per ml) coating for 4 h) or a polymer scaffold (Polycaprolactone nanofiber scaffold[23], 1 mg) was inserted. The solutions turned brown immediately, and then they were continued to incubate under 37 °C for 1 h. To remove the MnO₂ nanosheet with absorbed proteins, the solution was centrifuged for 3 times at 8801×g for 10 min, and precipitates were removed each time until there are no visible precipitates anymore. The solution should be transparent at this moment. 0.1 ml supernatant solution was transferred into a 96-well plate, and BCA (bicinchoninic acid assay, Thermo Fisher, A53226) was used to quantify the percentage of protein absorbed on nanosheets by subtracting the total amount of proteins remaining in the control group to the protein remained in the experimental groups. The assay was conducted strictly following the protocols from Thermo Fisher and absorption at 570 nm was used to quantify the protein amount for each group. These experiments were replicated for 3 times, and the values were normalized to the glass control. The amount of laminin absorbed on MnO₂ nanosheet was calculated by subtracting the laminin concentration after MnO₂ nanosheet absorption from the original concentration. The percentage of laminin absorption was calculated by dividing the amount of laminin absorbed by the original laminin concentration. BCA protein assay was repeated 3 times experimentally to get error bars shown in Fig. 1. Data are mean ± s.d., n = 3, **P < 0.01 by one-way ANOVA with Tukey post-hoc test.

**In vitro biodegradation of MnO₂ hybrid nanoscaffold.** We first studied the in vitro degradation of MnO₂ nanosheet in physiological conditions, different PBS solutions with varying concentrations of vitamin C (10 μg per ml, 50 μg per ml, 100 μg per ml, 200 μg per ml, 500 μg per ml) were prepared. Then 10 μl of 3 mg per ml MnO₂ nanosheet solution was added into 3 ml of vitamin C solution, and the UV-Vis spectrum of the solution was recorded every two minutes. The percentage of nanosheet remaining was normalized to the absorption (at 385 nm) of a control group without any vitamin C added. To study the degradation of thin layered nanoscaffold, we drop-casted 100 μL of MnO₂ nanosheet solution (1.0 g per ml) into the wells of 24 well plate treated with oxygen plasma. After vacuum drying for 3 h, a homogeneous, yellow and transparent film formed. Then the wells were coated with laminin (Thermo Fisher, Catlog No.: 23017015) and seeded with human neural stem cells at a cell density of 80k per well. The cells were cultured in standard differentiation media (w/o bFGF and no exogenous compounds) for different periods of time (1 day, 3 day, 7 day, 12 day, 17 day, and 22 day). The cells were then fully detached using acutase for 10 min at 37 °C. Followed by washing with PBS and de-ionized water, 24-well plate was vacuum dried. Based on the absorption of MnO₂ nanosheet at 385 nm, the degradation percentage was quantified by subtracting background (empty well) and normalize to the well without culturing the cells.

To study the degradation of thick layered nanoscaffold, a 3-layer, cell-MnO₂ nanoscaffold-cell sandwiched structure that mimic tissue structures was formed through centrifugation at 130×g in a 15 mL Eppendorf centrifuge tubes. A similar structure of GO nanoscaffold-cell construct was formed using the same protocols as a control. The first layer contains 1 million iPSC-NSCs. The second layer is composed of 1.0 mg of MnO₂ nanosheet or graphene oxide. The third layer (Top layer) was centrifuged down from another 1 million iPSC-NSCs. Degradation of the scaffolds was monitored by the volume change and thickness change of scaffold on a weekly base. Based on the assumption that the scaffold has identical radius and areas, the percentage of scaffold volume was normalized to the thickness that was measured on Day 1.

To demonstrate cell-seeded nanoscaffold can fast degrade under biocompatible redox conditions, an aqueous solution of 0.3 mg per ml MnO₂ nanosheet was filtered through a cellulose membrane, then a layer of dye-labeled (food dye) cells was formed on the MnO₂ nanosheet assembled substrate using a tri-circular PDMS (Polydimethylsiloxane, Dow Corning®) chamber. After the addition of 2.0 mg per ml ascorbic acid (Sigma-Aldrich) for 5 min, most of the dark color of MnO₂ nanosheet disappeared, and a layer pink colored cell layer was formed.

**Tunable biodegradation of MnO₂ nanoscaffolds.** To show the tunable biodegradability of MnO₂ nanoscaffolds, we controlled the geometrical and chemical structures of MnO₂ nanoscaffold (Supplementary Fig. 10) which includes: (i) thickness [0.2 H vs. 1 H (H = 0.4 mm), shown in a and b, respectively, which is achieved by filtrating different concentrations (0.6 mg per ml and 3.0 mg per ml, respectively) of MnO₂ nanosheet solution while keeping solution volume (1.0 ml) and filtrating area (1075 mm²) constant]; (ii) Height to surface area ratio (from 0.4 mm:1075 mm₂ in Supplementary Fig. 10b to 4 mm:107.5 mm² in Supplementary Fig. 10c), which is achieved by filtrating same amount of MnO₂ nanosheets (1.0 ml of 3.0 mg per ml) but reduced the filtrating area by 10 times; (iii) protein amount in the scaffold (MnO₂ nanosheets absorbed with 1.0 mg per ml vs. with 10 mg per ml bovine serum protein and then vacuum filtered). Degradation profile of different scaffolds obtained by measuring time-dependent manganese concentrations in the solution using Inductively coupled plasma mass spectrometry (ICP-MS, Fisons Instruments PlasmaQuad 2+). Each sample was measured 3 times to obtain error bars and standard deviation. To control scaffold degradation without any additional bioreductants, different cell densities (0, 0.1, 0.5, 1, and 5 million cells per well in a 24 well plate) were seeded onto MnO₂ nanoscaffold, and the complete degradation time was monitored by the full disappearance of dark color. 0.5 ml media was changed every two days. Each experiment was repeated twice. Results are summarized in Supplementary Fig. 10m.

**Fabrication and characterization of MnO₂ hybrid nanoscaffold.** To fabricate the MnO₂ hybrid nanoscaffolds, 10 ml of MnO₂ nanosheet solution at a specific concentration was filtered through a cellulose filter paper (pore size = 20 nm) under the vacuum condition. Then the filter paper was taken out and cut into sizes and shapes of choice. To transfer into a transparent glass substrate, cleaned glass was first treated with oxygen plasma, then the MnO₂ nanosheet deposited on filter paper was wetted with de-ionized water and pressed against the glass. A 2.0 kg per cm² pressure was placed on top of the filter paper for 8–12 h, and glass attached with MnO₂ nanosheet was detached from the weight. To remove the cellulose attached with nanoscaffold, the substrate was incubated in acetone for 0.5 h and then briefly washed in methanol for 1 h. The transparency of nanoscaffold can be easily tuned by using different concentrations of MnO₂ nanosheet solution. The 3 concentrations used in Supplementary Fig. 3 are 50 μg per ml, 100 μg per ml and 200 μg per ml. For cellular studies, the concentration of 200 μg per ml MnO₂ nanosheet solution was used throughout the study. Graphene oxide assembled scaffold was fabricated using an identical protocol with a graphene oxide aqueous solution of 200 μg per ml. FESEM (Field Emission Scanning Electron Microscopy, Zeiss with Oxford EDS) was used to characterize the nanoscaffold.

To form MnO₂ laminin hybrid nanoscaffold, 400 μL of MnO₂ nanosheet aqueous solution (2 mg per ml) was quickly added to 100 μL of laminin solution (1.0 mg per ml, PBS, PH = 7.4). Then the laminin conjugated MnO₂ nanosheet was centrifuged and re-suspended in 10 ml de-ionized water (MnO₂ nanosheet concentration was 80 μg per ml). After vacuum filtration, the cellulose filter paper or PCL fiber was cut into size and shape of interest for the following cell studies.

**HiPSC-NSC culture.** Human iPSC-NPCs were derived from human iPSCs (WT126 clone 8; and WT33 clone 1)[57]. iPSC-NPCs were expanded in a proliferation media containing DMEM/F12 with Glutamax (Invitrogen), B27-supplement (Invitrogen), N2 (Stem Cells), and 20 ng per mL FGF2 (Invitrogen). Tissue culture vessels were treated with Matrigel (Corning) 1:200 dilution with DMEM (Invitrogen) at 37 °C for 1 h. To initiate the neuronal differentiation

process, bFGF was removed. Fresh media was exchanged every other day. iPSC-NSCs with passage 8–11 were used in all our transplantation and in vitro studies.

**Differentiation of iPSC-NSC on MnO$_2$ nanoscaffold**. The viabilities of iPSC-NSCs and human neural progenitor cells (hNPC, Supplementary Methods) cultured on MnO$_2$ nanoscaffold were measured by presto blue cell viability assay (Thermo Fisher, Catalog No.: A13261, 10% volume ratio as compared to cell media). Into 24-well plates, laminin (10 µg/ml) was first coated onto the glass (control), graphene oxide assembled scaffold (positive control) and MnO$_2$ nanoscaffold at a concentration of 20 µg per ml (media volume: 1.0 ml per well), for 4 h. Then the iPSC-NSC was seeded into each well at a cell density of 20 k in growth media (bFGF added, 20 ng per ml). After the cells were cultured for 48 h, cell viabilities cultured on different substrates were quantified using fluorescence (excitation at 570 nm and emission at 590 nm) presto blue assay and normalized to the control group (glass). For the neurite length analysis, neurites on each substrate were first automatically traced and the lengths were automatically measured by NeuronJ in ImageJ software. The values are all averaged from 9–12 measurements in the representative immunostaining images[58]. Here is a summary of the average neurite lengths with standard deviation obtained from software: Glass control, 5.2 ± 2.3 µm; MnO$_2$ nanoscaffold, 59.3 ± 19.8 µm; MnO$_2$ laminin nanoscaffold, 84.5 ± 26.5 µm.

For the differentiation of iPSC-NSC on glass substrates, MnO$_2$ nanoscaffold, graphene oxide assembled scaffold (GO nanoscaffold) and glass were first sterilized in the ultraviolet (UV) lamp for 5 min and then coated with laminin solution (10 µg per ml) for 4 h. The substrates were placed in 24-well plates, and iPSC-NSCs were seeded into the wells at a cell density of 60 k cells per well. The cells proliferated for 24 h, and the media was changed to differentiation media without bFGF. To observe the stem cell proliferation and attachment onto the substrate, the cells were imaged in the optical microscope (Nikon Eclipse Ti-E microscope). After 6-days' differentiation, the cells were fixed and immunostained with nuclei (Hoechst, Thermo Fisher, catalog number: 33346, 1:100 dilution, 0.2 mM) and neuronal marker (TuJ1, Cell Signaling, catalog number: 4466, 1:500 dilution). To quantify the neuronal markers (TuJ1) and astrocyte markers (GFAP), qRT-PCR was conducted by using GAPDH mRNA as a control (Supplementary Table 2).

**Fabrication of MnO$_2$ laminin hybrid nanoscaffold**. MnO$_2$ laminin hybrid nanoscaffold can be facilely fabricated by adding 10 µL of MnO$_2$ nanosheet aqueous solution (3 mg per ml) into 100 µL laminin solution (1 mg per ml), and the MnO$_2$ nanosheet will be assembled within 5 s. To fabricate larger scale MnO$_2$ laminin hybrid nanoscaffold, 100 µL of MnO$_2$ nanosheet aqueous solution (3 mg per ml) was added into 500 µL laminin solution (1 mg per ml) and then vacuum filtered on a cellulose paper as described above. The structure of MnO$_2$ laminin hybrid nanoscaffold was then analyzed in FESEM. To fabricate cell encapsulated MnO$_2$ laminin-nanoscaffold, 1 million iPSC-NSCs were centrifuged down and re-dispersed in 25 µL laminin PBS solution. Different amount (0, 0.3, 1.5, 3, 15, and 30 µL) of MnO$_2$ nanosheet solution (3 mg per ml) was injected into the cell laminin solution, and a iPSC-NSC encapsulated pellet was spontaneously formed after one hour. To investigate the interaction between MnO$_2$ and encapsulated iPSC-NSCs, the medium was removed and the neurons were fixed in Formalin solution (Sigma-Aldrich) followed by two PBS washes. The biological samples (prepared under 15 µL MnO$_2$ condition) were then dehydrated to eliminate water through a series of ethanol dehydration process by replacing PBS with 50% ethanol/water, 70% ethanol/water, 85% ethanol/water, 95% ethanol/water, and absolute ethanol twice for 10 min each in succession. The biological samples were then stored in absolute ethanol before transferring to critical point dryer to eliminate traces of ethanol. Then 20 nm of gold was sputter coated onto the surface of biological samples after drying. FESEM was then used for micrograph acquisition.

For the differentiation of iPSC-NSC on substrates, glass, MnO$_2$ nanoscaffold and MnO$_2$ laminin hybrid nanoscaffold were first sterilized in the UV lamp for 5 min and then coated with laminin solution (10 µg per ml) for 4 h. The substrates were placed in 24-well plates, and iPSC-NSCs were seeded into the wells at a cell density of 60,000 per well. After 6 days' differentiation, the cells were fixed, and immunostaining on nuclei (DAPI) and neuronal marker (TuJ1) was conducted. Stem cell assays were repeated 3 times to obtain statistical information unless mentioned otherwise. Student t-test was used for two group analysis and ANOVA with Tukey post hoc test was used for multi-group (more than 3 groups) analysis. For the long-term (1 month, 30 days) stem cell differentiation assay, identical protocol was used with twice media change per week. Mature neuronal marker (MAP2) was used for identifying neurons differentiated from iSPC-NSCs on MnO$_2$ nanoscaffold.

Calcium imaging of neurons differentiated from iPSC-NSCs on MnO$_2$ laminin hybrid nanoscaffold in 12 well-plates. iPSC-NSCs were differentiated on MnO$_2$ laminin hybrid nanoscaffold using an identical protocol mentioned above for 6 days, then the cells were incubated with 1 ml of Fura-2 AM (Life Technologies, Catalog Number: F1201, 1:200 dilution, 5 µg per ml) in cell media for 1 h. Afterwards, cell media was changed to PBS. Under the movie mode of a fluorescence microscope, concentrated KCl solution in PBS (50 mM, 0.1 ml) was added to the cells, and the movie was taken for 10 min with 60 frames per seconds. The movies were pseudocolored, with red indicating strong calcium flux and green indicating weak calcium flux. An identical procedure was also applied for collecting calcium imaging of neurons differentiated from hNPCs. A summary of time dependent calcium intensity peaks can be found in Fig. 4, which was automatically obtained from the Nikon ND2 software.

**Dye loading on MnO$_2$ nanoscaffold and MRI studies**. To study drug loading and release on MnO$_2$ nanoscaffold, rhodamine B was used a model drug. Briefly, 0.3 mg rhodamine B (Alfa Aesar, Catalog Number: A13572) was added into 3.0 ml of MnO$_2$ nanosheet solution. After incubation at room temperature for 12 h, 5.0 ml PBS (PH = 7.4) was gradually added into the solution and RhB loaded MnO$_2$ nanosheet was centrifuged down at 3431 g for 5 min and extensively washed with PBS for 6 times to remove the residual RhB solution. Then the RhB-loaded MnO$_2$ nanosheet was re-suspended in 10 ml solution and re-assembled with laminin using the identical conditions for fabricating MnO$_2$ laminin hybrid nanoscaffolds. To monitor the dye hold-up, RhB-nanoscaffold was incubated with PBS for 12 h, then the fluorescence of the supernatant was detected by a fluorescence spectra (Varian Cary Eclipse). The dye loading was confirmed by degrading the RhB-nanoscaffold using 1.0 mg/ml ascorbic acid PBS solution. Instant appearance of pink color from RhB proves the loading of RhB inside nanoscaffold. RhB-nanoscaffold before and after degradation was also spotted in a glass slide in a close-proximity and then imaged in the fluorescent microscope. To test the correlation between MRI signals and RhB released, different amount of RhB-nanoscaffold [5, 2.5, 1, 0.5, 0.1 mg (from left to right in Fig. 5c)] were degraded with ascorbic acid (1.0 mg per ml) to form a homogeneous solution. Then the same solution in 96-well plates was used for MRI (Aspect's M2™ Compact High-Performance MRI, 1T) measurement and fluorescence measurement under Nikon fluorescent microscope.

To study the day-dependent drug (RhB) release from our MnO$_2$-laminin hybrid nanoscaffold, PBS with 10 µg per ml vitamin C was used to incubate the RhB loaded nanoscaffold, and was changed regularly every day. Fluorescence images (Nikon fluorescent microscope) were taken at Day1, Day2, and Day7, and the intensities from 3 different experiments were used to quantify the amount of RhB released. As a control, PCL polymer was dissolved with RhB and then formed a scaffold by drying at room temperature. Then the dye release was measured at the same time points as RhB loaded nanoscaffolds. The percentage of dye release was all normalized to the fluorescence intensity obtained at Day1.

To load neurogenic drugs into MnO$_2$ laminin hybrid nanoscaffold, DAPT (N-[(3,5-Difluorophenyl)acetyl]-L-alanyl-2-phenyl]glycine-1,1-dimethylethyl ester, Tocris, Catalog Number: 2634) was first dissolved in a PBS: DMF = 9:1 solution (dimethylformamide, Sigma-Aldrich) at a concentration of 0.1 mg per ml. Then 1.0 ml of DAPT solution was quickly mixed with 100 µL of 3 mg per ml MnO$_2$ nanosheet aqueous solution. After incubating for 12 h, the solution was centrifuged down and washed with de-ionized water for 3 times. The successful loading of DAPT onto MnO$_2$ nanosheet was confirmed by MALDI-TOF (Bruker, Ultraflex) based on the Na$^+$-DAPT peak at 455 (molecular weight to charge ratio). Briefly, 50 µL MnO$_2$ nanosheet loaded with DAPT solution was mixed with 50 µL gold nanoparticle solution (Ted Pella, 10 nm, $5.7 \times 10^{12}$ particles per ml). Then 1 µL of the mixed solution was drop-cast onto ITO glass and baked at 50 °C for 1 min to fully evaporate water. The DAPT solution was drop-cast on the same ITO glass as a reference. ITO glass was placed into the MALDI-TOF and exposed with a laser for the analysis.

To measure the DAPT loading and releasing profile on DAPT MnO$_2$ nanoscaffold, 3.0 mg MnO$_2$ nanosheets were first loaded with DAPT using the previous protocol and then assembled into MnO$_2$ laminin hybrid nanoscaffolds. Using UV-Vis spectroscopy, we first identified the characteristic absorption peak of DAPT (20 µM in water with 1% DMF) at 264 nm. Then the drug loading amount was determined by the full disappearance of the 264 nm peak after incubated with nanosheets (2.0 ml PBS, 1.5 mg per ml). This corresponds to a loading efficiency of 110 µg per 3.0 mg nanosheets and a molar ratio between DAPT and manganese atom = 1:134 (Supplementary Fig. 13). After that, to quantify the release of DAPT, degradation and DAPT release from the nanoscaffold was initiated by incubating it in a solution containing 10 µg per ml ascorbic acid. As DAPT form strong binding complex with MnO$_2$ nanosheets (Binding Energy = −18.43 kcal per mol), we monitored DAPT release through quantifying the amount of manganese amount at different time points, and estimated the drug release based on the constant molar ratio between DAPT and manganese (1:134) in the MnO$_2$-DAPT complex. The average daily release was quantified through dividing the total amount of manganese released by the length of degradation. The summarized DAPT release profile can be found in Supplementary Fig. 13.

**DFT calculations on small molecule and MnO$_2$ binding**. DFT calculations were carried out using the Quantum ESPRESSO software package. For the geometry optimization Perdew-Burke-Ernzerhof (PBE) functional along with D2 dispersion corrections were used[59,60]. The MnO$_2$ surface and the MnO$_2$ bound complexes were treated with DFT + U method. This is because conventional DFT functionals are unable to describe the strong correlation effect among the partially filled d states in Mn[61]. The Hubbard parameter 'U', is introduced for the Mn 3d electrons to describe the on-site Coulomb interaction, as given in the well-known GGA + U method[62]. The values of $U = 4$ eV and $J = 0$ eV for MnO$_2$ were adopted[63]. Spin-polarized calculations were performed since bulk MnO$_2$ has an antiferromagnetic ground state. The electron cores were defined using ultrasoft pseudopotential for all the elements and were extracted from the Quantum ESPRESSO main website (http://theossrv1.epfl.ch/Main/Pseudopotentials). For the k-point mesh, a γ-center

was used. The wave function cutoff of 60 Ry and kinetic energy cutoff of 240 Ry were used in all the cases studied. The Gaussian smearing was turned so that the difference between the free energy and the total energy is less than 0.005 Ry per atom. The energy convergence was set to $1 \times 10^{-6}$ a.u. and the force convergence threshold for the ionic minimization was set at $1 \times 10^{-4}$ a.u.

The binding energies on the $MnO_2$ surface were calculated for a series of small molecules (Supplementary Table 1) and the DAPT drug molecule. The size of the cell was taken equivalent to the size of the $MnO_2$ surface that has $8 \times 8$ oxygen atoms at the periphery. The box size for the simulated system is $23 \times 23 \times 40$ Å, and periodic boundary conditions are used. This condition was chosen to mimic the 2D $MnO_2$ surface. We first performed geometry optimizations for the bound complexes, with the resulting energy referred to as $E_{complex}$. We then optimized the structures of isolated $MnO_2$ surface and the molecule of interest, obtaining their energies $E_{MnO_2}$ and $E_{mol}$, respectively. The binding energy is defined as $E_b = E_{complex} - E_{MnO_2} - E_{mol}$. Negative $E_b$ indicates binding while positive $E_b$ indicates repulsion to the surface.

**In vivo transplantation of hiPSC-NSCs.** Spinal cord injury, transplantation of nanoscaffold and tissue harvest: All animal work was conducted following the regulation of the Institutional Animal Care and Use Committee (IACUC) at Rutgers University. The Notch1CR2-GFP transgenic mouse (*Mus musculus*) was used in this study. Adult mice that are 5–6 months old were picked for the spinal cord injury experiments. No difference was observed between male or female animals, and thus the gender was not specified. Animals are randomized without pre-knowledge of their behaviors and then assigned to different experimental groups without selection. Three mice were used for each group. Observers were blind to animal groups when performing experiments and analysis. During the surgery, initial anesthetization was performed with 5% isoflurane and then maintained at 2% Isoflurane. For hemisection, a laminectomy at T10~11 lateral section of spinal cord (length of gap ≈1 mm) was first performed. Then the dorsal blood vessel was burned with a cauterizer, and the spinal cord was cut from middle line toward the left using a #10 scalpel. Following induction of injury, bio-materials with iPSC-NSCs, laminin-coated and DAPT-treated PCL, or surgifoam was inserted into the wound site, and the muscle around surgical wound was sutured and skin is stapled using wound clips. A cell density of 1 million cells/cm$^2$ was used for transplantation and mice were sacrificed 7-week after injury. For harvest, the spinal cords from the injured animals were obtained via microsurgical dissection. They were washed in 1× PBS and fixed with 4% (w/v) paraformaldehyde (PFA) for 24 h. Fixed tissues were washed again and then cryopreserved in 30% (w/v) sucrose for 48 h. Afterwards, the spinal cord tissue was embedded in cryo-preserving media (Tissue Tek® OCT compound) and kept frozen at −80 °C. Spinal cord sections were stained with PH3, GFAP, and F4/80 antibodies to analyze the long-term effect of nanoscaffold on SCI (Fig. 8). One-way ANOVA was used for multi-group analysis. Data represents mean ± s.d., $n = 3$, *$P < 0.05$, **$P < 0.01$ with Tukey's post-hoc analysis. All tissue sections are in the center of the scaffold implants and then identified nearby the transplanted sites.

**In vivo hiPSC-NSC-GFP transplantation assay.** To study enhanced transplantation of iPSC-NSCs using our nanoscaffold, we further transplanted GFP labeled iPSC-NSC cells (iPSC-NSC-GFP) into a non GFP wild type C57BL/6 mouse strain[64]. To obtain GFP labeled cells, we transfected iPSC-NSCs with plasmid expressing EGFP using previously reported protocol (Supplementary Fig. 20). We confirmed the high transfection efficiency (>90%) and strong green fluorescence from iPSC-NSCs before seeded to the scaffold for in vivo cell transplantation using fluorescent microscope (Supplementary Fig. 20b). The surgical procedures for transplantation of iPSC-NSC-GFP into C57BL/6 were kept identical as the experiments on Notch1CR2-GFP mice and we repeated the immuno staining on tissue sections 1-week post transplantation and 1-month (30 days) post transplantation. In addition to the 3 groups [injury only (surgifoam insertion), 3D BHI nanoscaffold, and PCL cell group] evaluated in Notch1CR2 mice, we added 3 other important control groups to better support the therapeutic potential of our developed scaffold system: $MnO_2$ scaffold without laminin or DAPT but with cell transplantation ($MnO_2$ cell group), $MnO_2$ nanoscaffold with laminin and DAPT but without cell transplantation ($MnO_2$ DAPT group), and direct injection of GFP iPSC-NSCs with laminin (laminin cell group). All other conditions were kept identical to the experimental group (3D BHI nanoscaffold). Each group includes 3 animals ($n = 3$) to check reproducibility. We summarized the animal groups in Supplementary Fig. 20, and all the tissue analysis results were summarized in Fig. 7 and Supplementary Figs. 21–23. GFP, TuJ1 and MAP2 positive cells were first identified by automatic detection function in the NIS Nikon software (NIS element AR) then the amount of cells were recorded for making the graphs (Supplementary Methods). Percentage of Syn positive cells are quantified by first identify GFP + Syn + cells then divided by amount of GFP + cells in each section. Unpaired student *t*-test was used for two group significance analysis and one-way ANOVA was used for multi-group analysis. Data represents mean ± s.d., $n = 3$ unless described otherwise, *$P < 0.05$, **$P < 0.01$, ***$P < 0.001$. In Supplementary Fig. 20, when we count GFP + cells, we used the automatic detection function in the NIS Nikon software to identify GFP + cells shown in Supplementary Fig. 20e, then summarized the amount of GFP + cells at specific distance intervals (100 µm) in

the sagittal sections. All tissue sections are in the center of the scaffold implants and then identified nearby the transplanted sites.

**Data availability**. The data that support the findings of this study are available from the authors on reasonable request. New data related to this manuscript can also be found on the following link: https://figshare.com/projects/ A_Biodegradable_Hybrid_Inorganic_Nanoscaffold_for_Advanced_Stem_Cell_- Therapy/29040.

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

## Acknowledgements

We acknowledge financial support from the NIH Director's Innovator Award (1DP20D006462-01), NIH R01 (1R01DC016612-01), NIH R21 (1R21NS085569-01), New Jersey Commission on Spinal Cord (CSCR17IRG010, CSR13ERG005 (K.B.L.), and 15IRG006 (L.C.)), NSF CHE-1429062, CBET-1236508, American Chemical Society New Directions Award (PRF# 55869-ND10), and the University City Science Center's QED Award. M.P. and Y.L. acknowledge financial support from Graduate Training in Emerging Areas of Precision and Personalized Medicine (P200A150131) and NJCSCR [12FEL001], respectively. G.D. and L.W. acknowledge the Office of Advanced Research Computing at Rutgers for providing access to the Amarel cluster. The calculations have also used the Extreme Science and Engineering Discovery Environment supported by NSF (ACI-1548562). We would like to acknowledge Thanapat Pongkulapa, Prof. Gene Hall and Rutgers Molecular Imaging Center (RUMIC) for graciously helping us with PCR analysis, providing us access to the MALDI-TOF instruments and MRI imaging facilities respectively. We would also like to acknowledge Ning Chiang, Prof. Dongming Sun, and Prof. Wise Young for kindly providing us GFP labeled iPSC-NSCs for our in vivo experiments. Additionally, we are also grateful to Dr. Shreyas Shah, Brian Conley, and Kaixiong Tu for their valuable advices and scientific discussions during manuscript preparation.

## Author contributions

L.Y. and K.B.L. conceived and designed the experiments. S.T.D.C. and C.R. contributed to the cell studies and electron microscopy analysis. L.Y. performed the experiments. Y. L., M.P., and L.C. contributed to in vivo experiments and image analysis. G.D. and L. W. contributed to DFT calculations on drug-scaffold interaction. L.Y. and K.B.L analyzed the data. L.Y. and K.B.L wrote the manuscript. The principle investigator is K. B.L.

## Additional information

**Competing interests:** The authors declare no competing interests.

