## [Peer Review File · Nature Communications]

Reviewers' comments:

Reviewer #1 (Remarks to the Author):

Summary:

The article by Yang et al. describes the development of a biodegradable hybrid inorganic nanoscaffold (MnO₂) as a vehicle for stem cell-delivery to the injured spinal cord. The authors characterized the nanoscaffold in terms of drug loading and degradability, and demonstrated that it promoted neuronal differentiation of human and rodent cells when treated with laminin and DAPT. The combination of neural cells and nanoscaffold promoted greater functional recovery following transplantation into a mouse model of spinal cord injury.

Although the nanoscaffold and its properties were well described and the idea of promoting greater neuronal differentiation of the transplanted human neural cells with DAPT interesting, the study is not well presented, and consequently, it is confusing. Perhaps more importantly, some of the conclusions are inaccurate due to flaws in the methods. This prevents publication of the manuscript in its present form in any journal.

1. The authors state that they used NotchCR2-GFP mice. Why was this strain chosen? It is not optimal for xenografts and as stated in the cited paper (Tzatzalos et al., 2012), not all cells are GFP-positive. Therefore counting any GFP-negative cell as transplanted is inaccurate for the analysis and leads to wrong conclusions.

- a. The analyses should be repeated using double labelling with a human specific maker.
- b. The full reference for Tzatzalos et al. should be included.

2. The nanoscaffold with laminin and DAPT on its own might promote axonal regeneration / neuronal differentiation of endogenous stem cells. It should be included in the analysis and compared to the cell-seeded and a no laminin, no DAPT nanoscaffold control. The PCL scaffold data could be omitted, but not the sham control.

3. Fig.4c: The time points investigated should be indicated.

4. What concentration of DAPT was bound to the nanoscaffold and what was the release profile?

5. On page 9, the authors state that the hybrid nanoscaffold system can be degraded in vitro and in vivo, but the in vivo data was not demonstrated at that point.

- a. Did laminin influence the degradation rate of the nanoscaffold?

6. Fig. SI19a: the label anti-human nuclei is in red, but the red staining is rather filamentous and not nuclear.

7. All methods used should be clearly described, e.g.:

- a. All antibodies used should be listed with the concentrations used.
- b. How was the neurite length analyzed? With an automated program?
- c. The primer sequence for the human beta III tubulin should be double checked as the listed sequence does not encode for it, but for MLLT11. In addition, both GAPDH primers are labelled human.
- d. Was part of the cord resected for implantation of the biomaterial? If yes, how long was the gap created?

8. The addition of rodent neural stem cells seems to add little to the story and could be omitted.

Reviewer #2 (Remarks to the Author):

The use of biodegradable MnO₂ sheets as biomaterial carriers for delivery of therapeutic stem cells and drugs is an innovative and intriguing idea. However, several of the claims made seem to be over-inflated given the experimental design and data presented.

1) Controlling material properties:

a. The authors argue that a major advantage of the MnO₂-laminin scaffolds for SCI is their rapid degradation. However, it is not clear why this would be the case. In fact, others have posited that many commonly investigated scaffolds (e.g., collagen- or fibrin-based) degrade too quickly so that the scaffold material is gone before axons have time to bridge the injury (e.g., Uibo, et al. (2009) *Biochim Biophys Acta* 1793:924).

b. It is also not clear that degradation, and thus drug release, can be controlled as stated by the authors. The data show that in vitro either changing the number of MnO₂ sheets or the amount of reducing agents can alter degradation rate. However, it is unlikely that the redox environment in vivo can be controlled and/or predicted well enough to tune degradation. If the number of MnO₂ sheets is used to control degradation – what are the limits? i.e., how many sheets maximum can be included and implanted to extend degradation times? How precisely can the degradation rate be tuned using this method?

c. The “summary and outlook” section describes that the mechanical and magnetic properties of MnO₂-laminin scaffolds can also be controlled; however, there do not appear to be data or references to previous reports demonstrating this claim.

2) Analysis of neuronal differentiation:

a. The authors use TUJ1 as a marker to show definitive differentiation into neurons. TUJ1 is considered an early marker of neuronal differentiation, but does not indicate a mature neuron. It should be clarified that these cells have begun to progress towards a neuronal lineage, but does not necessarily mean they will become mature neurons. Mature neuronal markers such as NeuN would be much better to demonstrate this.

b. It is not clear from only Ca²⁺ imaging that functional neurons are present. First, glial cells and even NSCs can be depolarized so that Ca²⁺ flux is present. At a minimum, it should be demonstrated Ca²⁺ flux overlaps with TUJ1⁺ cells. The presence of mature neurons could also be demonstrated by staining for proteins found in neuronal synapses.

c. In animal experiments, it is not clear that the TUJ1⁺ cells are from the transplanted NSCs. Was a human specific antibody used? If not, a double stain for human-specific nuclei (as used in the supplementary figures 18 and 19) would be required to confirm.

d. It is not clear from the images in Fig. 5 that more TUJ1⁺ cells derived from implanted NSCs are present in the 3D-BHI scaffolds than PCL or sham conditions. Was any quantification done to confirm in vivo observations? It would also help to include comparisons to controls at equivalent times. In particular, TUJ1 staining is shown at day 7 for 3D-BHI, but day 1 for PCL and sham in Fig. 5. No data are provided comparing differentiation or cell survival at days 0 and days 7 for all conditions.

3) SCI model:

a. It is not clear if the hemisection model is dorsal or lateral. The supplementary methods imply that it is lateral, but this needs to be stated explicitly.

b. Is GFAP staining shown in Fig. 5 at the same time point (week 7) for all conditions? GFAP changes dramatically in the first two weeks after injury, so it is crucial to compare conditions at the same time point to assess astrocyte reactivity. Was any quantification done to measure scar thickness?

c. Although GFAP is a great indicator of astrocyte reactivity, reduction of GFAP expression does not insure biocompatibility. Immune cell presence (at least numbers of microglia/macrophages) is another important measure.

d. It is not clear why PCL would be used as a “positive” control. One would not expect PCL to perform better than a scaffold containing laminin, which has been widely shown to enhance neurite outgrowth and neuronal differentiation. Better controls might be laminin alone and/or MNO2 sheets alone, which would enable decoupling of the effects of each scaffold component.

e. The data do not really show that NSC survival is improved in the BHI scaffolds. There is no quantification of surviving cells. The images do appear to indicate that there are more proliferating cells in BHI scaffolds; however, this is also not quantified. From the data presented, it is possible that equal numbers of NSCs survive transplantation, but BHI scaffolds promote proliferation. However, the authors state that BHI scaffolds improve survival and proliferation.

f. How do the authors reconcile that scaffolds promote both proliferation and differentiation of NSCs? Typically, if differentiation increases, proliferation decreases.

g. Although the BMS results are promising, a score of 6 at week 7 after injury has been reported with several other combinations of scaffolds, drugs and/or cells. Why is this strategy better than previous strategies?

4) Minor issues:

a. Figures c2 and c3 would be improved by including white scale bars.

b. The legend is missing for the graph in S1Fig7b.

I. SUMMARY OF THE KEY CHANGES THAT WERE MADE IN THE MANUSCRIPT TO ADDRESS THE REVIEW COMMENTS

Reviewer #1	Revised figures/texts	Reviewer #2	Revised figures/texts
1a	Revised FIG.5	1a	New SI FIG.11
1a	New SI FIG.20, 21, 22, 23	1b	New SI FIG.11
1a	New FIG. 6	1c	Page 17-21, Line 298-331
1a	Page 17-21, Line 298-331	2a-b	New SI FIG.7
1a	Added SI Methods A.18	2c-d	New FIG.6, SI FIG.20-24
1b	Page 24, Line 449-450	2d	Revised FIG.5
2a	New SI FIG.21	3a-b	Revised FIG.5
2b	New FIG.6, SI FIG.20-25	3b	New SI FIG. 24
3	FIG. 4 legend	3c	Revised FIG.5
4	New SI FIG.14	3d	New FIG.6, SI FIG. 20-23
5	Page 9, Line 174	3e	New SI FIG. 23 , Revised SI FIG. 19
5a	New SI FIG.11	3f	Explained in response letter
6	New SI FIG.20-23	3g	Explained in response letter
7a	SI methods A.7, A.11, A.20	4a	FIG.3 c2 and c3
7b	SI methods A.11	4b	Deleted SI Fig-7
7c	SI TAB.2		
7d	SI methods A.18		
8	Deleted SI Fig-7		

II. DETAILED RESPONSE TO THE REVIEWERS' COMMENTS AND THE CORRESPONDING REVISIONS THAT WERE MADE

We would like to thank the reviewers for their constructive comments. By supplementing new experiments, providing additional controls and rephrasing the manuscript, we believe we addressed all the concerns from reviewers and improved the quality and clarity of the manuscript significantly. Here we list our point-by-point responses with additional experimental evidence, references and text changes:

1. RESPONSE TO SPECIFIC COMMENTS MADE BY REVIEWER 1:

The article by Yang et al. describes the development of a biodegradable hybrid inorganic nanoscaffold (MnO₂) as a vehicle for stem cell-delivery to the injured spinal cord. The authors characterized the nanoscaffold in terms of drug loading and degradability and demonstrated that it promoted neuronal differentiation of human and rodent cells when treated with laminin and DAPT. The combination of neural cells and nanoscaffold promoted greater functional recovery following transplantation into a mouse model of spinal cord injury.

Although the nanoscaffold and its properties were well described and the idea of promoting greater neuronal differentiation of the transplanted human neural cells with DAPT interesting, the study is not well presented, and consequently, it is confusing. Perhaps more importantly, some of the conclusions are inaccurate due to flaws in the methods.

RESPONSE: We appreciate the reviewer's comments and suggestions. By providing new data, updating references and rephrasing the manuscript, we believe that we substantially improved the clarity of the manuscript and the conclusions.

COMMENT 1-1: *The authors state that they used Notch1CR2-GFP mice. Why was this strain chosen? It is not optimal for xenografts and as stated in the cited paper (Tzatzalos et al., 2012), not all cells are GFP-positive.*

Therefore counting any GFP-negative cell as transplanted is inaccurate for the analysis and leads to wrong conclusions.

RESPONSE 1-1: We thank the reviewer for bringing up this important issue. Notch1CR2-GFP mice were animal models designed for studying activation of endogenous neural stem cells after central nervous system injuries. For this study, Notch1CR2-GFP mice were used as an indirect way of determining the effectiveness of transplanted cells. In adult Notch1CR2-GFP mice, there is not a significant level of Notch expression; and we did not detect high levels of green fluorescence in healthy adult mice in our previous studies. However, after inducing injury to the spinal cord and after transplanting non-GFP human iPSC-NSCs in an animal with spinal cord injury, we could observe higher levels of GFP from the host cells from spinal cord comparing to transplanted cells surrounding the scaffold. As such, we utilized this observation for identifying transplanted cells. Although Notch pathway is not prevalent, a low level of Notch signaling is required to maintain NSC activity in the adult central nervous system. After the injury, endogenous NSCs were activated, which leads to the higher expression of Notch1 and its accompanied GFP fluorescence. We acknowledge that utilizing such an indirect method to determine the effectiveness of transplanted cells is not well-evaluated.

To address this issue, we performed new *in vivo* experiments on a wild-type C57BL/6 mouse strain and utilized GFP labeled iPSC-NSCs for xenografts (updated **Figure 5-6**, updated **SI Figure 20-23**). C57BL/6 mice is a mouse strain commonly used for xenografts¹. In fact, Notch1CR2-GFP mice are modified from C57BL/6, so they should share the same genetic background, behavior characteristics and responses to injuries to a large extent. Among different methods to identify transplanted cells in xenografts, GFP labeling is an accurate and optimal method for scaffold-based cell transplantation studies². This is particularly the case when we found anti-human nuclei do not work reliably.

To obtain GFP labeled cells, we transfected iPSC-NSCs with lentiviral vectors expressing GFP (**SI Figure 20**). We confirmed the high transfection efficiency (>90%, **SI Figure 20b**) and strong green fluorescence from iPSC-NSCs before seeded to the scaffold for *in vivo* cell transplantation (**SI Figure 20 e-f**). The surgical procedures were kept identical to our previous experiments on Notch1CR2-GFP mice, and we repeated the immunohistological staining on tissue sections. With the GFP labeled iPSC-NSCs, we can now accurately analyze the transplanted cells. We found that 3D-BHI nanoscaffold enhances survival and neuronal differentiation of the transplanted iPSC-NSCs as compared with other experimental and control groups (updated **Figure 6**).

In addition to the 3 original groups (control groups with injury only, nanoscaffold group with cell transplantation and PCL scaffold with cell transplantation), we evaluated in our previous manuscript, we added 3 more animal groups as important control groups in our new experiments to better support the therapeutic potential of our newly developed scaffold system: MnO₂ scaffold without laminin or DAPT but with cell transplantation (MnO₂ cell group), MnO₂ nanoscaffold with laminin and DAPT but without cell

transplantation (MnO₂ DAPT group), and direct injection of GFP-iPSC-NSCs with laminin (cell laminin group). Each group was repeated on 2 animals (total mice number: 12). Consistent with our previous study on Notch1CR2-GFP mice, all animals were sacrificed one-week post-injury, and immunostaining was performed under identical conditions. We summarized the animal groups in **SI Figure 20**, and all the tissue analysis results were summarized in updated **Figure 6** and **SI Figure 21-22**. Now with the GFP labeled cells, we can clearly observe the distribution of transplanted cells, and reliably perform dual-labeling experiments to study the fates of transplanted cells (**SI Figure 20f**). Therefore, by substantially optimizing the *in vivo* experiment design and providing new data, we believe the accuracy of our conclusions has been significantly improved.

COMMENT 1-2: *The full reference for et al. should be included.*

RESPONSE 1-2: We thank the reviewer for this kind suggestion. Now we included this reference in our updated manuscript (Page 24, Line 449, updated reference #53).

COMMENT 2-1: *The nanoscaffold with laminin and DAPT on its own might promote axonal regeneration / neuronal differentiation of endogenous stem cells. It should be included in the analysis and compared to the cell-seeded and a no laminin, no DAPT nanoscaffold control.*

RESPONSE 2-1: DAPT is a Notch inhibitor that selectively guides neural stem cell differentiation into neurons but not astrocytes. Its beneficial role in enhancing neural stem cell differentiation in spinal cord injury has also been previously shown³. Laminin coated scaffolds have also been applied to spinal cord injury by replacing inhibitory ECM post-injury. Therefore, nanoscaffold with laminin and DAPT may promote neuronal differentiation of endogenous stem cells. To address this concern, we supplemented two additional experimental groups: i) a cell-seeded nanoscaffold without laminin or DAPT control (MnO₂ cell group); ii) transplantation of MnO₂ nanoscaffold with laminin and DAPT without GFP labeled iPSC-NSCs (MnO₂ DAPT group). GFP labeled iPSC-NSCs and C57BL/6 mice were used as cell source and mouse strain for xenografts, respectively. 1 week later, the mice were sacrificed, and tissue sections were analyzed by immunostaining on neuronal (TuJ1) and astroglial (GFAP) markers. The immunostaining results were summarized in **SI Figure 21** and **SI Figure 22**.

We first evaluated the effects of different experimental groups on neurogenesis of endogenous neural stem cells (eNSCs) post-injury. Newly formed neuronal cells derived from eNSCs but the not transplanted cells (GFP+ cells) or from the injured axons (no nuclei should be existent) were identified based on the amount of DAPI+/TuJ1+/GFP- cells. From the representative immunostaining images (**SI Figure 21**), all three groups with DAPT and laminin, including nanoscaffold group (N=76), PCL cell group (N=49) and MnO₂ DAPT group (N=103) showed significant higher DAPI+/TuJ1+/GFP- cell counts compared to the control group (N=19), whereas simply injecting cells with laminin or transplanting cells by MnO₂ scaffold without therapeutic reagents do not seem to have an obvious effect on increasing neuronal cells derived from eNSCs. Future studies still need to be done to elucidate whether this increased neuronal populations on our nanoscaffold were originated from laminin or the therapeutic effects of DAPT; and whether the transplantation of our DAPT and laminin functionalized scaffolds induced more eNSCs migration into the injured area that increased the total amount of TuJ1+ cells. However, our immunohistological staining clearly indicates the beneficial role of our nanoscaffold treated condition on eNSCs differentiation into neurons as compared to direct cell injection with laminin.

New SI Figure 21. 3D-BHI nanoscaffold enhances neuronal differentiation. Immunohistological staining was performed to determine the effects on eNSC neurogenesis from different experimental treatments. A high number (N) of eNSC derived neuronal cells was observed in nanoscaffold group and MnO₂ DAPT group. Scale bar: 100 μ m.

In addition to studying the therapeutic effects on neuronal cell population from eNSCs, we also investigated the influence of scaffold transplantation on astroglial cell intensities at the injury sites (SI Figure 22). From the summarized graph, our nanoscaffold condition reduced astroglial cell levels at the injury sites significantly (58% reduction compared to control and 23% compared to MnO₂ DAPT group). Astroglial cells accumulating in the injured sites have been identified as a major reason for impeded axonal growth and functional recovery after spinal cord injury^{4,5}. Our nanoscaffold condition that has both iPSC-NSCs transplantation, which can secrete tropic factors, and release DAPT for reducing astrogliogenesis of eNSCs could be the reasons. This reduced astroglial marker intensities 1-week post injury also well correlate our long-term (7-week post injury) studies on reduced astroglial scar formation and the promoted functional recovery (Figure 5, SI Fig. 24) after nanoscaffold transplantation.

COMMENT 2-2: *The PCL scaffold data could be omitted, but not the sham control.*

RESPONSE 2-2: The main idea of our manuscript is to develop a new type of nanoscaffold for stem cell therapy and demonstrate its potential benefits as compared to conventional glass or polymer-based scaffolds. Therefore, we selected PCL scaffold, which has been widely applied for tissue engineering as well demonstrated for transplanting neural stem cells into injured spinal cords⁶. It is important to note that, both our nanoscaffold and PCL scaffold were loaded with DAPT and laminin under the identical conditions and concentrations. This way, the key variable between these two scaffold systems would be the materials, which allow us to be assured of the superior performance from our newly developed MnO₂ scaffold for stem cell transplantation.

We agree that analysis of sham control is important and should be included. As we switched the mouse strain (C57BL/6) for our experimental group and control group, we also repeated our sham condition using identical surgical procedures and analyzed the endogenous neuronal cells and astroglial cells by using immunohistological staining (SI Figure 20-25).

COMMENT 3: Fig.4c: The time points investigated should be indicated.

RESPONSE 3: We thank the input from the reviewer. **Figure 4c** is used for supporting our idea of predicting drug releasing amount using MRI signals. We used 5 different groups with varying amount model drug (rhodamine B) loaded scaffold, then degraded them using bioreductants (vitamin C) over an identical period (2 days, SI section A.15.). The solution was then collected for the analysis. Therefore, Fig. 4c is not a time-dependent study. To avoid this confusion, we stated more clearly in the main text underneath **Figure 4**. On the other hand, in case the readers were interested in the time-dependent release of rhodamine B, we included the drug releasing profile in **SI Figure 13**.

COMMENT 4: What concentration of DAPT was bound to the nanoscaffold and what was the release profile?

RESPONSE 4: The loading concentration of DAPT to the nanoscaffold is 22 $\mu\text{g}/600 \mu\text{g}$ (over 90% of DAPT was absorbed onto the scaffold, which corresponds to 3.6% weight percentage of drug/scaffold ratio; the molar ratio of DAPT: Mn=1:134). This characterization is performed using UV-Vis absorption spectroscopy (**SI Figure 14a**) by using the characteristic peak of DAPT at 264 nm in a co-solvent of water/DMF. We have also quantified the average daily DAPT release DAPT for 2 weeks (volume=5.0 ml), which is summarized in **SI Figure 14b**. The average daily DAPT release varies from 0.6 μM to 3.2 μM from Day1 to Day10, which is within therapeutic window of DAPT for guiding neurogenesis. Unlike Rhodamine B, DAPT has a slower daily release initially, which can be due to the higher hydrophobicity of DAPT.

COMMENT 5-1: On page 9, the authors state that the hybrid nanoscaffold system can be degraded *in vitro* and *in vivo*, but the *in vivo* data was not demonstrated at that point.

RESPONSE 5-1: We appreciate the reviewer's suggestion. Even though we showed the *in vivo* degradation in the later sections (**SI Figure 17**), we agree that mentioning in the earlier sections where no data was shown at that time point can be confusing to the readers. Therefore, we removed the term of "*in vivo*" in that sentence.

COMMENT 5-2: Did laminin influence the degradation rate of the nanoscaffold?

RESPONSE 5-2: To study the effect of laminin on the degradation rate of the nanoscaffold, we prepared laminin-containing and no laminin nanoscaffolds using the identical amount of MnO_2 nanosheets. We incubated both scaffolds under identical concentrations of ascorbic acid (10 $\mu\text{g}/\text{ml}$) to initialize the degradation. Concentrations of manganese in the solutions were collected at different time points and measured using ICP-MS. The summarized degradation profiles can be found in **SI Figure 11**, together with other means of tuning scaffold degradation. From the graph, we concluded that the incorporation of laminin leads to a slow-down of degradation

by 2 times. This could be due to the strong interactions between laminin and MnO₂ nanosheets, which can bond nanosheets tightly and reduce the reactions between an ascorbic acid and MnO₂ nanosheets in the scaffold.

COMMENT 6: *Fig. S119a: the label anti-human nuclei is in red, but the red staining is rather filamentous and not nuclear.*

RESPONSE 6: We did occasionally observe this abnormal staining results in our study. As we were not able to solve this issue by repeating the anti-human nuclei staining, we have switched our labeling method by performing new *in vivo* experiment using GFP labeled iPSC-NSCs for xenograft. Compared to anti-human nuclei staining or our previously used counter-staining method in Notch1CR2 GFP mice, GFP labeling is a more direct and reliable way to identify transplanted cells. The characterization of our viral transfected GFP-iPSC-NSC line was shown in **SI Figure 20**. We also removed the figures with anti-human nuclei staining.

COMMENT 7: *All methods used should be clearly described, e.g.:*

RESPONSE 7: We have updated accordingly to describe methods clearly.

COMMENT 7-1: *All antibodies used should be listed with the concentrations used.*

RESPONSE 7-1: We have updated the dilution factors and concentrations in the SI methods part.

COMMENT 7-2: *How was the neurite length analyzed? With an automated program?*

RESPONSE 7: We thank the reviewer for suggesting an automated program to measure neurite lengths. In this study, neurites from iPSC-NSCs differentiated on nanoscaffold and control scaffolds were first identified by manually tracing neurites using TuJ1 immunostained cells. The lengths were then measured using Nikon NIS Elements software. The average neurite length was averaged from 5 measurements.

As this method of neurite measurement is still based on manual tracing, we also repeated the analysis on neurite length using NeuronJ plugin in Image J software⁷, and here is a summary for the average neurite lengths: Glass control, 5.2+2.3 μm; MnO₂ nanoscaffold, 59.3+19.8 μm; MnO₂ laminin nanoscaffold, 84.5+26.5 μm; DAPT loaded BHI nanoscaffold, 125+30.2 μm (n=9-12 for each group). This trend well matches manually traced neurite lengths. As the automated program is a more objective method for quantifying neurite lengths, we replaced the values in the manuscript **Figure 1, 3** and **4**. We also updated the methods part in the SI accordingly.

COMMENT 7-3: *The primer sequence for the human beta III tubulin should be double checked as the listed sequence does not encode for it, but for MLLT11. In addition, both GAPDH primers are labeled human.*

RESPONSE 7-3: We are grateful for the reviewer's comments. Now we have corrected the primer sequence and the GAPDH primer labeling accordingly. Primer sequence for human beta III tubulin: Forward:

Targets (species)	Forward Primer (5' to 3')	Reverse Primer (5' to 3')
GAPDH (Human)	CCGCATCTTCTTTGCGTCCG	GCCCAATACGACCAAATCCGT
TuJ1 (Human)	GGTGTCCGAGTACCAGCAGT	TTCGTACATCTCGCCCTCTT
GFAP (Human)	AGGAAGATTGAGTCGCTGGA	AACCTCCTCCTCGTGGATCT
GAP43 (Human)	AACCTGAGGCTGACCAAGAA	GGGACTTCAGAGTGGAGCTG
FAK (Human)	TGGTGAATGGAGCGAGTATT	CAGTGAACCTCCTCTGACCG

GGTGTCCGAGTACCAGCAGT; Reverse: TTCGTACATCTCGCCCTCTT. We do not include the primers for rat species as rodent studies were removed as suggested in Comment #8.

COMMENT 7-4: *Was part of the cord resected for implantation of the biomaterial? If yes, how long was the*

gap created?

RESPONSE 7-4: We performed hemisection, where a cut was made on the spinal cord followed by the insertion of surgifoam to keep the gap constant. The gap is typically 1 mm. We have also updated in the methods part (A.18) accordingly.

COMMENT 8: *The addition of rodent neural stem cells seems to add little to the story and could be omitted.*

RESPONSE 8: Our original intention to test the therapeutic effects of our new nanoscaffold on a different stem cell line may expand its broad applicability. However, as we only used iPSC-NSCs for the rest of our manuscript, it could be distracting to readers, and we now omitted this supporting figure to avoid confusion.

2. RESPONSE TO SPECIFIC COMMENTS MADE BY REVIEWER 2:

The use of biodegradable MnO₂ sheets as biomaterial carriers for delivery of therapeutic stem cells and drugs is an innovative and intriguing idea. However, several of the claims made seem to be over-inflated given the experimental design and data presented.

RESPONSE We appreciate the reviewer's support and valuable comments. By re-designing the experiments and providing new data, now we significantly improved the strength of conclusions.

COMMENT 1-1: *Controlling material properties:*

The authors argue that a major advantage of the MnO₂-laminin scaffolds for SCI is their rapid degradation. However, it is not clear why this would be the case. In fact, others have posited that many commonly investigated scaffolds (e.g., collagen- or fibrin-based) degrade too quickly so that the scaffold material is gone before axons have time to bridge the injury (e.g., Uibo, et al. (2009) Biochim Biophys Acta 1793:924).

RESPONSE 1-1: We thank the reviewer for bringing up this discussion. It is correct that fast degradable collagen or fibrin-based scaffolds may not be beneficial to bridge the gap in the spinal cord injury and provide longer-term support for axons to grow through. On the other hand, when scaffolds were applied for cell transplantations, a slow biodegradability is known to restrict cell migration, proliferation, and lead to nutrient and oxygen deficiencies for cells, in which case fast degradation is desired^{8,9}. In fact, one of the recently used and highly recognized scaffolds that successfully induces neuronal differentiation of iPSC-NSCs *in vivo* and long-distance axonal growth of differentiated neurons is based on a fast degradable fibrin scaffold². This gives us a good inspiration for designing our fast biodegradable MnO₂ nanoscaffolds. Despite these considerations, the highly heterogeneous microenvironments and complex scaffold properties make it less likely to determine the most optimal degradation speed for all scaffold based cell transplantations to treat different kinds (e.g. contusion model vs. hemisection model) of spinal cord injury. Therefore, it is perhaps more obvious that if a scaffold can have a wide and reliable tunability over biodegradation speed, such scaffold could be advantageous as its biodegradability can be well optimized for specific applications^{8,9}. To this end, we supplemented new data showing the effective and wide-range control of our MnO₂ nanoscaffold biodegradation speed through multiple methods (**SI Figure 11**). Briefly, our nanoscaffold can be controlled to degrade rapidly with full degradation in a week, or slowly – around 20% degradation after 2 weeks. This wide range tunable and therapeutic relevant degradation profile make us believe the MnO₂ nanoscaffold represents a promising candidate for transplanting stem cells to treat not only central nervous system injuries but also apply to the broad tissue engineering applications.

COMMENT 1-2: *It is also not clear that degradation, and thus drug release, can be controlled as stated by the authors. The data show that in vitro either changing the number of MnO₂ sheets or the amount of reducing agents can alter degradation rate. However, it is unlikely that the redox environment in vivo can be controlled and/or predicted well enough to tune degradation. If the number of MnO₂ sheets is used to control degradation – what are the limits? i.e., how many sheets maximum can be included and implanted to extend degradation times? How precisely can the degradation rate be tuned using this method?*

RESPONSE 1-2: We appreciate the reviewer's suggestion on controlling the scaffold degradation through tuning scaffold properties other than redox. To address this comment, we modulated the geometrical and chemical structures of MnO₂ nanoscaffold (SI Figure 11 a-e) which includes: i) thickness (0.2 H vs. 1 H, H=0.4 mm, shown in a and b, respectively, which is achieved by filtrating different concentrations of MnO₂ nanosheet solution while keeping solution volume and filtrating area constant); ii) aspect ratio (height to surface area ratio, shown in b and c, which is achieved by filtrating same amount of MnO₂ nanosheets but reduce the filtrating area by 10 times); iii) protein amount in the scaffold (MnO₂ nanosheets absorbed with 1.0 mg/ml vs. with 10 mg/ml). Redox conditions (ascorbic acid concentration of 10 μg/ml) were kept constant and physiologically relevant for all the conditions.

We precisely monitor the degradation profile by measuring time-dependent concentrations of manganese elements that dissolved in the solution through ICP-MS. Percentage of degradation at each time point was normalized to the total amount of manganese existent in the nanoscaffolds prepared. From the summarized graph (SI Figure 11 g), all degradation profiles show zero-order degradation kinetics for most of the time, and we can clearly conclude the wide range tunability of scaffold biodegradation by comparing different profiles. Briefly, reducing the thickness of nanoscaffold by 5 times can increase the degradation speed by about 3 times; increasing the aspect ratio while maintaining MnO₂ nanosheets the same slow down the degradation speed by over 10 times; and increasing the protein concentrations utilized to assemble nanosheets lead to a significant increase around 7 times. Overall, the degradation of MnO₂ nanoscaffold can be controlled from fast (full degradation within 3 days) to slow (around 30% degradation after 2 weeks), which cover the wide ranges desirable for different tissue engineering applications.

New SI Figure 11. Tunable biodegradability of MnO₂ nanoscaffolds by modulating scaffold structure and varying cell densities. a-f, 6 different methods to prepare scaffolds and for modulating scaffold degradation. Symbols were kept consistent with Scheme in manuscript. Green fibers in d and e represent bovine serum protein. b represents the control scaffold to show the tuning of biodegradability by comparing it to a, and c-f. g, Degradation profile of different scaffolds obtained from measuring time-dependent manganese concentrations in the solution using ICP-MS. h-m, Tuning scaffold degradation by controlling cell amount transplanted on the scaffold. h-k, Schematics showing scaffolds seeded with varying amount of iPSC-NSCs. m, A summary of full degradation time of scaffolds based on the complete disappearance of scaffold color. MnO₂ laminin nanoscaffold was used for all the conditions, and regular iPSC-NSC differentiation media was used for maintaining cell viabilities with regular daily change. n=3 and 2 in g and m, respectively. Error bars are standard error of the mean.

In addition to tuning scaffold structure, we were also curious about how to modulate scaffold degradation in the absence of any exogenous trigger such as ascorbic acid. To this end, we also transplanted different amounts of iPSC-NSCs (N=0, 0.5, 1.0 and 5.0 million, SI Figure 11 h-k) to achieve such controllable degradation. iPSC-

NSC differentiation media without any additional bioreductants was used for this study, and we monitored the full degradation of scaffold based on the complete disappearance of brown colored MnO₂ nanosheets. Summarized full degradation time can be found in **SI Figure 11m**. As expected, when no cells were seeded onto the scaffold, no noticeable degradation from our MnO₂ nanoscaffold happened throughout the one-month period of observation. With cell transplanted, the degradation time shows a clear cell density-dependent trends: transplanting 5 million iPSC-NSCs can lead to scaffold degradation within 2 days, while 0.5 million cells will sustain for over 2 weeks. As such, we can conclude that we can reliably control scaffold degradation independent of redox, and thus drug releasing using varying approaches that modulate scaffold structures.

COMMENT 1-3: *The "summary and outlook" section describes that the mechanical and magnetic properties of MnO₂-laminin scaffolds can also be controlled; however, there do not appear to be data or references to previous reports demonstrating this claim.*

RESPONSE 1-3: We thank the reviewer for this suggestion. When describing magnetic properties, we were talking about the MRI imaging and drug release monitoring from the released Mn²⁺. We have also successfully tuned the mechanical properties of our MnO₂ nanoscaffold by controlling its porosities (data not shown), which is currently under study in our lab. However, as it can be confusing for readers if we do not show these data at this moment; therefore, we removed the claim of tuning mechanical properties in the conclusion part.

COMMENT 2-1: *Analysis of neuronal differentiation:*

The authors use TUJ1 as a marker to show definitive differentiation into neurons. TUJ1 is consider an early marker of neuronal differentiation, but does not indicate a mature neuron. It should be clarified that these cells have begun to progress towards a neuronal lineage, but does not necessarily mean they will become mature neurons. Mature neuronal markers such as NeuN would be much better to demonstrate this.

RESPONSE 2-1: We appreciate the reviewer's comments and valuable suggestion. TuJ1 is an early neuronal marker and was used in our study. Meanwhile, it is important to investigate more mature neuronal markers that

could indicate their potential role on signal relaying. To this end, we repeated our differentiation experiments under identical conditions on our nanoscaffold, fixed cells at Day 6 and performed immunostaining using MAP2 and Synapsin 1 antibodies. From our summarized **SI Figure 7**, we can conclude that a majority of TuJ1 positive cells differentiated on our nanoscaffold are co-labeled with MAP2 and Synapsin1. MAP2 protein belongs microtubule-associated protein family that is enriched in dendrites; Synapsin is a neuronal marker associated with functional maturation of neuronal synapses. Both proteins are well-known markers for characterizing mature neurons^{2,10}. Therefore, with these supplemented immunostaining images using mature neuronal

New SI Figure 7. iPSC-NSCs differentiated on MnO₂ laminin nanoscaffolds expresses mature neuronal markers (MAP2 and Synapsin 1). Shown in this figure is immunostaining images of iPSC-NSCs differentiated for 6 days.

markers such as MAP2 and Synapsin, now we can better demonstrate iPSC-NSCs will become mature neurons.

COMMENT 2-2: *It is not clear from only Ca²⁺ imaging that functional neurons are present. First, glial cells and even NSCs can be depolarized so that Ca²⁺ flux is present. At a minimum, it should be demonstrated Ca²⁺ flux overlaps with TUJ1+ cells. The presence of mature neurons could also be demonstrated by staining for proteins found in neuronal synapses.*

RESPONSE 2-2: We thank the reviewer for his/her suggestions. As glial cells and even NSCs can be depolarized and evoke fast and local Ca²⁺ elevations during astrocytic processes, calcium imaging alone may not be sufficient to indicate mature neurons. Therefore, we performed immunostaining on differentiated neurons using mature neuronal markers (MAP2 and Synapsin), which is summarized in **SI Figure 7**. By staining differentiated cells with proteins found in neuronal synapses (Synapsin), we can demonstrate the presence of mature neurons.

COMMENT 2-3: *In animal experiments, it is not clear that the TUJ1+ cells are from the transplanted NSCs. Was a human specific antibody used? If not, a double stain for human-specific nuclei (as used in the supplementary figures 18 and 19) would be required to confirm.*

RESPONSE 2-3: This is related to Reviewer #1 Comment #1a. After scaffold transplantation into the spinal cord, both transplanted iPSC-NSCs and endogenous neural stem cells (eNSCs) can differentiate into TuJ1+ cells. To demonstrate the potential of our nanoscaffold, it is important to identify transplanted iPSC-NSCs from eNSCs. As the human-specific antibody does not work reliably for us, and utilizing Notch1CR2 GFP mice based counterstain is an indirect way to identify transplanted cells, we re-designed our *in vivo* experiments on a wild-type (non-GFP labeled) C57BL/6 mouse strain, and utilized GFP labeled iPSC-NSCs for xenografts (updated **Figure 6, SI Figure 20-23**). To identify transplanted cells in xenografts, GFP labeling is a widely used method and optimal for scaffold-based cell transplantation studies⁶. In addition to the 3 main groups (control groups with injury only, nanoscaffold group with cell transplantation and PCL cell

group with cell transplantation) that we evaluated in our previous manuscript, we added 3 new animal groups as important control groups to better support the therapeutic potential of our newly developed scaffold system: MnO₂ scaffold without laminin or DAPT but with cell transplantation, MnO₂ nanoscaffold with laminin and DAPT but without cell transplantation, and direct injection of GFP iPSC-NSCs with laminin. To obtain GFP labeled cells, we transfected iPSC-NSCs with lentiviral vectors expressing GFP (SI Figure 20). We repeated the immunohistological staining on tissue sections. With the GFP labeled iPSC-NSCs, we can now accurately analyze the transplanted cells through GFP+/TuJ1+ dual labeling. Summarized data can be found in SI Figure 20-23. From updated Figure 6 and through counting TuJ1+/GFP+ cells using dual labeling, now we can confirm the significantly (3 times increase) higher amount of MnO₂ nanoscaffold transplanted iPSC-NSCs (3D-BHI nanoscaffold group) become neurons as compared to the other control groups. The guided differentiation of iPSC-NSCs on our nanoscaffold is further evidenced by a minimum amount (1 out of 9) of astroglial cells differentiated from our transplanted iPSC-NSCs through counting GFAP+/GFP+ cells (SI Figure 20).

COMMENT 2-4: *It is not clear from the images in Fig. 5 that more TUJ1+ cells derived from implanted NSCs are present in the 3D-BHI scaffolds than PCL or sham conditions.*

RESPONSE 2-4: As summarized in Figure 6, now we can conclude that more (10 times higher) TuJ1+ cells derived from implanted NSCs are present in the 3D-BHI scaffolds than PCL conditions. This could originate from a combined therapeutic effect from enhanced iPSC-NSCs graft, which can be observed from higher populations of GFP+ cells existent in the injured sites, and improved neurogenesis efficiency from nanoscaffold condition compared to PCL based transplantation.

COMMENT 2-5: *Was any quantification done to confirm in vivo observations?*

RESPONSE 2-5: We thank the reviewer for this suggestion. Now in our updated Figure 6 and SI Figures (SI Figure 20-24), we performed quantifications for all the immunohistological staining (GFP, GFAP, TuJ1, cleaved Caspase3, PH3) that we have performed.

COMMENT 2-6: *It would also help to include comparisons to controls at equivalent times. In particular, TUJ1 staining is shown at day 7 for 3D-BHI, but day 1 for PCL and sham in Fig. 5. No data are provided comparing differentiation or cell survival at days 0 and days 7 for all conditions.*

RESPONSE 2-6: We appreciate the reviewer's comments. Now by re-designing the *in vivo* experiment, we included comparisons to controls that include both injury-only group and 3 new control groups to elucidate the therapeutic effects at equivalent times (1-week post-injury). We also provide data comparing differentiation (DAPI, GFP, GFAP, TuJ1 staining, Figure 6, SI Figure 20-22) for all conditions (6 experimental groups) and cell survival (Caspase 3 staining, SI Figure 23) for the nanoscaffold condition and PCL cell group at Day 7.

COMMENT 3-1: *SCI model:*

It is not clear if the hemisection model is dorsal or lateral. The supplementary methods imply that it is lateral, but this needs to be stated explicitly.

RESPONSE 3-1: To avoid this confusion, we included the description of lateral hemisection model in our updated Figure 5 and A.18 in the SI Methods part.

COMMENT 3-2: *Is GFAP staining shown in Fig. 5 at the same time point (week 7) for all conditions? GFAP changes dramatically in the first two weeks after injury, so it is crucial to compare conditions at the same time point to assess astrocyte reactivity.*

RESPONSE 3-2: GFAP staining shown in Figure 5 are all at the same time point of Week 7. To make it clearer, we updated Figure 5 by labeling each image with detailed time points.

COMMENT 3-3: Was any quantification done to measure scar thickness?

RESPONSE 3-3: We appreciate the reviewer for suggesting the quantification of glial scars, and we included this analysis in **SI Figure 24**.

COMMENT 3-4: Although GFAP is a great indicator of astrocyte reactivity, reduction of GFAP expression does not insure biocompatibility. Immune cell presence (at least numbers of microglia/macrophages) is another important measure.

RESPONSE 3-4: Immune cell presence after transplantation of scaffolds with cells is important to identify biocompatibility and is also known to influence functional recovery after injury. To address this comment, we characterized the immune cells (mainly macrophages) using F4/80 antibodies. Longer time points (7-week post injury) were studied, as inflammation in the injured sites has been stabilized. From our summarized

graph (updated **Figure 5f**), interestingly, we observed even lower macrophage accumulation nearby both scaffolds (3D-BHI nanoscaffold and PCL scaffold) as compared to no transplantation group. This can be due to the transplantation of iPSC-NSCs which are known to secrete trophic factors for suppressing immune reactions

¹¹. Even though the detailed mechanism of the reduced immune cell presence remains to be studied, this result, combined with its beneficial effects on animal behaviors (functional recovery), indicate a good biocompatibility of our 3D-BHI nanoscaffolds.

COMMENT 3-5: It is not clear why PCL would be used as a “positive” control. One would not expect PCL to perform better than a scaffold containing laminin, which has been widely shown to enhance neurite outgrowth and neuronal differentiation.

RESPONSE 3-5: This is a similar question raised by Reviewer #1 Comment #2b, and we appreciate both reviewer’s comments. The main idea of our manuscript is to develop a new type of nanoscaffold for stem cell therapy and demonstrate its potential benefits as compared to conventional glass or polymer-based scaffolds. Therefore, we selected PCL scaffold, which has been widely applied for tissue engineering as well demonstrated for transplanting neural stem cells into injured spinal cords ⁶. It is important to note that, both our nanoscaffold and PCL scaffold were loaded with DAPT and laminin under the identical conditions and concentrations. This way, the key variable between these two scaffold systems would be the materials, which allow us to be assured of the superior performance from our newly developed MnO₂ scaffold for stem cell transplantation. Additionally, as it is confusing to interpret PCL as a positive control, we changed it to PCL control instead.

COMMENT 3-6: *Better controls might be laminin alone and/or MnO₂ sheets alone, which would enable decoupling of the effects of each scaffold component.*

RESPONSE 3-6: We thank the reviewer for suggesting 2 suitable control groups. As described in the scheme in **SI Figure 20**, now we included controls with cell injection with laminin, and MnO₂ scaffold (with DAPT loaded and laminin functionalized) alone, which would enable decoupling the effects of each scaffold component. We compared the effects of each group on both endogenous neural stem cells and transplanted GFP-iPSC-NSCs at the identical timepoint of 7-day post-injury. Compared to cell injection with laminin group, our 3D-BHI nanoscaffold condition not only enhanced the amount of GFP-iPSC-NSC retention significantly (by over 10 times, **Figure 6b**), promoted higher percentages of iPSC-NSC neurogenesis (22% higher, based on the percentage of TuJ1+/GFP+ cells, **SI Figure 6c**), but also improved the neuronal populations from eNSCs (based on the counts of DAPI+/TuJ1+/GFP- cells, **SI Figure 21**). Compared to MnO₂ scaffold alone with DAPT and laminin functionalized but without cell transplantation, our 3D-BHI nanoscaffold treated group (with iPSC-NSC transplantation) reduce the GFAP signal intensities nearby the injury site by a larger margin (by 24%, based on GFAP immunohistological staining, **SI Figure 22**).

COMMENT 3-7: *The data do not really show that NSC survival is improved in the BHI scaffolds.*

RESPONSE 3-7: We appreciate the reviewer's suggestions. To support that survival of transplanted cells is improved in the 3D-BHI nanoscaffold, we performed immunohistological staining 1-week post injury by using cleaved Caspase 3 antibodies. We quantified the percentage of Caspase 3+/GFP+ cells over GFP+ cells and GFP-iPSC-NSCs transplanted by PCL scaffold were used as a control. From the summarized **SI Figure 23**, we found that the percentage of cells that are undergoing apoptosis (indicated by Caspase 3 staining) is significantly lower (2-fold decrease) in nanoscaffold treated animals as compared to the PCL cell group. In fact, as can be seen in **SI Figure 21**, we found much even more differences regarding amounts of GFP+ cells remained 1-week after transplantation (over 10 times higher in nanoscaffold group comparing to PCL cell group). These results collectively support the cell survival is improved in the 3D-BHI (cell transplanted and drug-loaded MnO₂) nanoscaffolds.

COMMENT 3-8: *The images do appear to indicate that there are more proliferating cells in BHI scaffolds; however, this is also not quantified.*

RESPONSE 3-8: We appreciate the reviewer's comments. Now we performed the quantification and updated it in **SI Figure 19**.

COMMENT 3-9: *However, the authors state that BHI scaffolds improve survival and proliferation.*

RESPONSE 3-9: Now with the immunohistological staining results using cleaved Caspase 3 antibodies, we can support this statement.

COMMENT 3-10: *How do the authors reconcile that scaffolds promote both proliferation and differentiation of NSCs? Typically, if differentiation increases, proliferation decreases.*

RESPONSE 3-10: While many pathways such as Notch inhibitors promote differentiation through inhibiting proliferation, there are several well-known pathways (e.g., WNT) and therapeutic reagents (e.g., lithium treatment for GSK- β 3 inhibition) that are known to enhance both proliferation and differentiation of NSCs or neural progenitor cells¹²⁻¹⁴. In our case, it could be due to the upregulated laminin densities on our nanoscaffold that promote both proliferation and differentiation of NSCs. Laminin has been known to be essential for and promote both the proliferation of neural stem cells and their differentiation *in vitro* and *in vivo*^{15,16}. In the injured sites after spinal cord injury, the laminin-rich extracellular matrix is often replaced by inhibitory molecules (e.g., chondroitin sulfate proteoglycan) that impedes neuronal differentiation and do not favor NSC proliferation⁴. Therefore, by providing a favorable extracellular matrix for our transplanted iPSC-NSCs, both proliferation and differentiation of NSCs could be promoted.

COMMENT 3-11: *Although the BMS results are promising, a score of 6 at week 7 after injury has been reported with several other combinations of scaffolds, drugs and/or cells. Why is this strategy better than previous strategies?*

RESPONSE 3-11: We thank the reviewer for bringing up this discussion. Even though many efforts have been made, currently there is no effective cure established for spinal cord injury. While BMS score is a good indicator of functional recovery, they can vary significantly based on the injury model (e.g., hemisection, transection, and contusion models), levels of injury (e.g., depth of cut and weight used for contusion) and surgeon's experience. Therefore, it is usually difficult to control all these factors identical when comparing BMS score across different laboratories and different experimental conditions. In our experiment, by comparing nanoscaffold treated conditions to a laminin-coated polymer scaffold that previously evaluated for stem cell transplantation, we still achieve obvious enhancement on functional recovery, which could support the potential of our nanoscaffold for stem cell transplantation and therapies. By providing more detailed cell imaging on other important control groups (e.g., cell injection with laminin, MnO₂ nanoscaffold alone without cells), the behind mechanisms of our nanoscaffold treated condition for promoting functional recovery were further investigated. While there is still room for us to further optimize scaffold properties (e.g., degradation speed, injectability, DAPT loading amount, mechanical properties, or combing with anti-inflammatory drugs) to achieve higher BMS scores, our biodegradable and biocompatible nanoscaffold system that enhances cell survival, guides neurogenesis of iPSC-NSCs *in vitro* and *in vivo* and deliver therapeutic reagents locally could represent a unique and promising candidate for current scaffold systems for stem cell-based treatment for spinal cord injury. Additionally, our current study is still largely focused on developing our new hybrid nanoscaffold systems for stem cell therapy and tissue engineering in general, rather than providing ultimate cures for spinal cord injury. By demonstrating its attractive points regarding high biocompatibility, tunable biodegradability, guiding stem cell differentiation, delivering the drug in a controlled manner and providing imaging modalities, it could also be applied for other stem cell-based applications.

COMMENT 4: *Minor issues:*

a. Figures c2 and c3 would be improved by including white scale bars.

b. The legend is missing for the graph in SIFig7b.

RESPONSE 4: We appreciate the reviewer for carefully proof-reading the figures. Now we have updated the scale bars in **Figure 3 c2** and **c3**. We deleted SI Figure 7 based on the comments from Reviewer #1 Comment #8.

III. REFERENCES CITED FOR THIS RESPONSE LETTER:

- 1 Luchetti, S. *et al.* Comparison of immunopathology and locomotor recovery in C57BL/6, BUB/BnJ, and NOD-SCID mice after contusion spinal cord injury. *Journal of neurotrauma* **27**, 411-421, (2010).
- 2 Lu, P. *et al.* Long-distance growth and connectivity of neural stem cells after severe spinal cord injury. *Cell* **150**, 1264-1273, (2012).
- 3 Dias, T. B., Yang, Y.-J., Ogai, K., Becker, T. & Becker, C. G. Notch signaling controls generation of motor neurons in the lesioned spinal cord of adult zebrafish. *Journal of Neuroscience* **32**, 3245-3252, (2012).
- 4 Thuret, S., Moon, L. D. & Gage, F. H. Therapeutic interventions after spinal cord injury. *Nature Reviews Neuroscience* **7**, 628-643, (2006).
- 5 Yiu, G. & He, Z. Glial inhibition of CNS axon regeneration. *Nature Reviews Neuroscience* **7**, 617-627, (2006).
- 6 Wang, T.-Y., Forsythe, J. S., Nisbet, D. R. & Parish, C. L. Promoting engraftment of transplanted neural stem cells/progenitors using biofunctionalised electrospun scaffolds. *Biomaterials* **33**, 9188-9197, (2012).
- 7 Meijering, E. *et al.* Design and validation of a tool for neurite tracing and analysis in fluorescence microscopy images. *Cytometry Part A* **58**, 167-176, (2004).
- 8 Kraehenbuehl, T. P., Langer, R. & Ferreira, L. S. Three-dimensional biomaterials for the study of human pluripotent stem cells. *Nature methods* **8**, 731-736, (2011).
- 9 Loh, Q. L. & Choong, C. Three-dimensional scaffolds for tissue engineering applications: role of porosity and pore size. *Tissue Engineering Part B: Reviews* **19**, 485-502, (2013).
- 10 Ambasudhan, R. *et al.* Direct reprogramming of adult human fibroblasts to functional neurons under defined conditions. *Cell stem cell* **9**, 113-118, (2011).
- 11 Lee, S.-T. *et al.* Anti-inflammatory mechanism of intravascular neural stem cell transplantation in haemorrhagic stroke. *Brain* **131**, 616-629, (2007).
- 12 Moon, R. T., Kohn, A. D., De Ferrari, G. V. & Kaykas, A. WNT and β -catenin signalling: diseases and therapies. *Nature Reviews Genetics* **5**, 691-701, (2004).
- 13 Su, H., Chu, T.-H. & Wu, W. Lithium enhances proliferation and neuronal differentiation of neural progenitor cells in vitro and after transplantation into the adult rat spinal cord. *Experimental neurology* **206**, 296-307, (2007).
- 14 Chen, B. Y. *et al.* Brain-derived neurotrophic factor stimulates proliferation and differentiation of neural stem cells, possibly by triggering the Wnt/ β -catenin signaling pathway. *Journal of neuroscience research* **91**, 30-41, (2013).
- 15 Song, H., Stevens, C. F. & Gage, F. H. Astroglia induce neurogenesis from adult neural stem cells. *Nature* **417**, 39-44, (2002).
- 16 Hall, P. E., Lathia, J. D. & Caldwell, M. A. Laminin enhances the growth of human neural stem cells in defined culture media. *BMC neuroscience* **9**, 71, (2008).

Reviewers' comments:

Reviewer #1 (Remarks to the Author):

The authors made extensive changes to the manuscript, which improved it in my opinion. However, the manuscript continues to lack clarity and should be more concise (including supplementary materials and figure legends). Important points that need improving in regards to cell therapies, long term cell survival and fate, are not investigated. Cell survival and cell fate were only analysed at an early time point (new figure 6). Others have shown that controlling early cell fate, e.g. by pre-differentiation, does not necessarily mean that the cell population remains differentiated into the desired cell type in the long-term (e.g. DOI: 10.1088/1748-605X/aa96dc). The statistical analysis used should be mentioned in the methods and significances indicated in the graphs.

Other points for further consideration include:

1. Abbreviations should be introduced at their first use and then used throughout the text / figures (e.g. weeks post injury - wpi).
2. Indicate the time point of sacrifice clearly in the figure legend (e.g. Fig. SI 21).
3. A-6: How long were the iPS-NSC cultured for? FGF2 can inhibit oligodendroglial differentiation and early passages produce more neurons compared to later ones.
"Similar to the rNSC protocol" – this part was taken out and should not be referred to.
4. It should be mentioned in the figure legend of Fig. 4C that samples were incubated for 2 days.
5. The sentence in lines 204-206 needs to be fixed.

Summary:

The article by Yang et al. describes the development of a biodegradable hybrid inorganic nanoscaffold (MnO₂) as a vehicle for stem cell-delivery to the injured spinal cord. The authors characterized the nanoscaffold in terms of drug loading and degradability, and demonstrated that it promoted neuronal differentiation of human and rodent cells when treated with laminin and DAPT. The combination of neural cells and nanoscaffold promoted greater functional recovery following transplantation into a mouse model of spinal cord injury.

Although the nanoscaffold and its properties were well described and the idea of promoting greater neuronal differentiation of the transplanted human neural cells with DAPT interesting, the study is not well presented, and consequently, it is confusing. Perhaps more importantly, some of the conclusions are inaccurate due to flaws in the methods. This prevents publication of the manuscript in its present form in any journal.

1. The authors state that they used NotchCR2-GFP mice. Why was this strain chosen? It is not optimal for xenografts and as stated in the cited paper (Tzatzalos et al., 2012), not all cells are GFP-positive. Therefore counting any GFP-negative cell as transplanted is inaccurate for the analysis and leads to wrong conclusions.
 - a. The analyses should be repeated using double labelling with a human specific maker.
 - b. The full reference for Tzatzalos et al. should be included.

2. The nanoscaffold with laminin and DAPT on its own might promote axonal regeneration / neuronal differentiation of endogenous stem cells. It should be included in the analysis and compared to the cell-seeded and a no laminin, no DAPT nanoscaffold control. The PCL scaffold data could be omitted, but not the sham control.
3. Fig.4c: The time points investigated should be indicated.
4. What concentration of DAPT was bound to the nanoscaffold and what was the release profile.
5. On page 9, the authors state that the hybrid nanoscaffold system can be degraded in vitro and in vivo, but the in vivo data was not demonstrated at that point.
 - a. Did laminin influence the degradation rate of the nanoscaffold?
6. Fig. SI19a: the label anti-human nuclei is in red, but the red staining is rather filamentous and not nuclear.
7. All methods used should be clearly described, e.g.:
 - a. All antibodies used should be listed with the concentrations used.
 - b. How was the neurite length analyzed? With an automated program?
 - c. The primer sequence for the human beta III tubulin should be double checked as the listed sequence does not encode for it, but for MLLT11. In addition, both GAPDH primers are labelled human.
 - d. Was part of the cord resected for implantation of the biomaterial? If yes, how long was the gap created?
8. The addition of rodent neural stem cells seems to add little to the story and could be omitted.

Reviewer #2 (Remarks to the Author):

Thank you for thoroughly addressing the previous comments. However, a few concerns remain.

The statistical methods used to assess significance must be included in text – preferably in the appropriate figure captions and supplementary methods.

In a few places (e.g., Introduction lines 77, 78), the authors argue that redox-tunable biodegradable is an advantage as stem cell scaffolds for SCI therapy. Although it is true that this property should make the scaffolds readily degrade in an inflammatory environment, it is not clear that this is an advantage for clinical applications – where inflammation may vary widely – or for delivery of neurogenic compounds such as DAPT – which ideally would be delivered over longer times to affect stem cell differentiation. Similarly, how would degradation of these materials be controlled clinically using scaffold thickness? It seems like altering scaffold size is not ideal for clinical settings – where patient size and injury type may determine what scaffold size is required.

There are a few instances where the authors described BHI scaffolds as having “enhanced binding affinity toward ECM proteins” (e.g., line 103). Enhanced protein adsorption compared to what? The data show glass as a negative control, which proteins often adsorb poorly to and is not therapeutically

relevant. For laminin coating to be effective, glass is typically first etched and then often coated with polylysine before adding laminin. It does not appear like this was done for laminin-coated glass controls here and thus it is difficult to discern whether changes in the conformation of laminin adsorbed to glass versus nanosheets or simply the concentration of laminin is responsible for differences in cell adhesion, spreading and differentiation. On a similar note, how does adsorption to MnO₂ nanosheets affect laminin structure? Many of the sites in laminin known to bind cell receptors rich in aromatic and amine groups (e.g., IKVAV and YIGSR). Finally, what type of laminin was used (was it laminin I or another form?)?

The authors should be more cautious when claiming that the transplanted cells really become “functional” neurons after only 7 weeks in vivo. While they may have some electrical activity (Ca²⁺ imaging) and immature synapses (synapsin protein) there is no evidence that these cells have integrated (or are capable of integrating) with functional circuits. Even the BMS data showing functional recovery could easily be due to production of anti-inflammatory and regenerative factors by the stem cells, rather than their incorporation into host circuitry.

For the in vivo data shown in Figure 5, the captions states that “Error bars are the standard error of the mean (n=2)”, implying that 2 separate animal cohorts were evaluated. However, how many animals were included in each cohort? Similarly, for counts of GFP+ cells (as in SI Fig. 20), was one section counted per region (SI Fig. 20f)? For Figure 6, the legend says that only 3 sections were counted per animal. Were these all in the same region (e.g., center of the scaffold implant)? What about for other in vivo quantification data (e.g., Caspase3, PH3)? Also, how were cells counted? Manually or by stereology? This method should be described.

Since biodegradation of scaffolds was assessed by scaffold thickness (line 165), does this imply that scaffolds degrade by surface erosion and not bulk degradation? If this is the case, do nutrients/water for embedded cells get in between the nanosheets, just not through the nanosheets?

In the Summary (line 348), the authors state that our developed biodegradable hybrid inorganic nanoscaffold-based stem cell therapeutic approach would be a useful tool for selectively controlling stem cell differentiation and neuronal behaviors in vivo”. While it may be able to do this with future refinements, in the current form the data only show good survival of transplanted cells and the capacity of these cells to become neurons. The data do not show that differentiation and behavior can be selectively controlled.

Minor comments:

In the Methods, the authors describe using GFP-encoding lentivirus to “transfect” iPSC-NSCs prior to transplanted. However, the correct term would be “transduce” or “infect”. “Transfect” would be used to describe plasmid delivery.

The images in Fig. 2e are difficult to see.

Abstract (line 49) – The wording of this sentence (“To this end, we developed a novel biodegradable nanoscaffold-based method for the controlled delivery of therapeutic molecules, and improved spinal cord injury (SCI) treatment through enhanced stem cell therapy by improving stem cell survival, neuronal differentiation, and neurite outgrowth in a mouse model.”) is oddly redundant and really not clear.

In the Introduction (line 63), the authors broadly refer to “stem cell therapy”. However, this should read “stem cell transplants” in the context of the rest of the sentence so that therapies aimed at

targeting endogenous stem cells are excluded.

In the Introduction (line 66), please include a citation to support the claim that most cells die shortly after transplantation.

Introduction (line 68) – “Therefore” can be deleted.

Introduction (lines 80-81) – The authors state that BHI scaffolds “have extraordinary properties” but do not elaborate on these in this section so that the sentence reads like “filler” with no real content.

Introduction (line 88) – “applications” should be “application”

Lines 294-297 – It seems that stem cell survival may reduce glial scar formation (through production of anti-inflammatory factors) and stem cell differentiation may help re-establish neuronal circuits. In its current form, the text implies that both cell survival and differentiation may prevent glial scar formation, which has not really been shown.

We would like to thank both reviewers for their constructive comments. We are also excited to know reviews appreciated the significance, impacts, innovation of our research works. According to the suggestions of reviewers and by supplementing new experimental data and rephrasing the manuscript accordingly, we further improved the clarity and flow of the revised manuscript. Herein we listed our point-by-point responses with additional experimental evidence, references and text changes (Please note that **changes in main texts (revised manuscript) were highlighted using yellow highlight color in the manuscript and supporting files**; updated figures were also heightened by **red dashed-outlines**):

RESPONSE TO SPECIFIC COMMENTS MADE BY REVIEWER 1:

The authors made extensive changes to the manuscript, which improved it in my opinion. However, the manuscript continues to lack clarity and should be more concise (including supplementary materials and figure legends).

RESPONSE: We appreciate the reviewer's strong support and constructive comments. To follow up them, We have revised the manuscript, supplementary materials, and figure legends accordingly to further improve manuscript clarity.

Less important methods and supporting figures were also removed to **Figshare** (https://figshare.com/projects/A_Biodegradable_Hybrid_Inorganic_Nanoscaffold_for_Advanced_Stem_Cell_T_therapy/29040) to make manuscript and related files more concise.

COMMENT 1-1: *Important points that need improving in regards to cell therapies, long term cell survival and fate, are not investigated. Cell survival and cell fate were only analysed at an early time point (new figure 6). Others have shown that controlling early cell fate, e.g. by pre-differentiation, does not necessarily mean that the cell population remains differentiated into the desired cell type in the long-term (e.g. DOI: 10.1088/1748-605X/aa96dc).*

RESPONSE 1-1: We appreciate the reviewer's comments. It is an important goal in the field of neural stem cell transplantation to achieve long-term (1 month) survival and controlled differentiation in clinical relevant settings¹. Currently, most successful attempts in this regard using human neural stem cell transplantation have been achieved through the delivery of complex growth factor cocktails using immunodeficient animal models, which may have clinical translation concerns². To address the comment "*long-term cell survival and fate, are not investigated.*", we first performed *in vitro* stem cell differentiation assay on 3D BHI nanoscaffold (DAPT-loaded MnO₂ laminin hybrid nanoscaffold) (new **SI Fig. 16**). After one month, cells were fixed, and immunostaining was performed. To identify neurons, a mature neuronal marker MAP2 was used. Based on the immunostaining results from this long-term study, large populations of cells

New SI Figure 16. Long-term neuronal differentiation of iPSC-NSCs on 3D BHI nanoscaffold. Immunostaining was performed 1 month (30 days) post transplantation. Blue indicates nuclei staining and green indicates mature neuronal marker MAP2. These results clearly support that large populations of differentiated cells and their neuronal fates remained after long-term (1-month) differentiation process on 3D BHI nanoscaffold.

remain on the scaffold, suggesting a good cellular adhesion on our 3D BHI nanoscaffold during the 1-month period. More importantly, most of the cells still express mature neuronal marker and have long neurite extensions. In addition, these neurons differentiated from iPSC-NSCs form a semi-3D network-like structure (new SI Fig. 16). These results are consistent with our short-term *in vitro* studies previously performed and strongly support the potential of our new nanoscaffold for long-term induction of iPSC-NSC neuronal differentiation.

After confirming the potency of 3D BHI nanoscaffold for long-term induction of neuronal differentiation *in vitro*, we also performed new *in vivo* experiments and transplanted iPSC-NSC-GFP using our 3D-BHI nanoscaffold (experimental condition). As the control condition, iPSC-NSC-GFP was also directly injected. Three animals were used for both experimental and control condition, and all animals were sacrificed one month (30 days) after transplantation. Surgical procedures and tissue staining protocols were the same as previously described for short-term (1WPI) *in vivo* studies. The fates of transplanted iPSC-NSC-GFP cells were examined by early (TuJ1) and mature (MAP2 and Synapsin) neuronal markers (New SI Fig. 25). In control animals, GFP+ cells were barely found in the cell injection condition. This is consistent with our previous 1WPI *in vivo* experiments (Fig. 6) and could be ascribed to highly unfavorable microenvironments as well as the immune response from the wild-type animals.

In contrast, while there is also a decrease on GFP+ cell counts and reduced neuronal populations comparing to 1 WPI condition (Fig. 6), our 3D BHI nanoscaffold treated condition (SI Fig. 25) showed a significantly higher amount of GFP+ cells at the sites of transplantation when comparing to the control condition (* $P < 0.05$). More importantly, most of the GFP+ cells express early neuronal marker (e.g., TuJ1) and some cells express mature neuronal markers (e.g., MAP2 and synapsin). Previous studies that used cocktails of growth factor also achieved robust neuronal differentiation and functional recovery in the long term, but such demonstration in wild-type animals with a simple formulation has been less successful². Collectively, we

New SI Fig. 25. Enhanced transplantation and neuronal differentiation of iPSC-NSCs by 3D BHI nanoscaffold 1-month post injury. **a**, Schematic diagram illustrating the transplantation of iPSC-NSC-GFP into injured spinal cord for one month. **b-c**, Statistic summary (**b**) and corresponding immunostaining images (**c**) from 3D-BHI nanoscaffold group and cell injection group 1-month post-injury (in the 3rd graph, 3 columns represent percentage of TuJ1+, Synapsin+ and MAP2+ cells existent in the GFP+ cells, respectively). All tissue sections in **c** were from the center of spinal cord injury and then selected nearby transplantation sites. GFP, TuJ1 and MAP2 positive cells were counted in individual sections (674 μm by 674 μm). Data in **b** are mean \pm s.d., $n=3$, * $P < 0.05$ by unpaired student t-test.

conclude that the enhanced functional recovery is due to i) improved neuronal differentiation and re-establishment of the signal relaying; ii) improved cell survival. These lead to more significant secretion of neurotropic factors by the stem cells. Thus, our *in vitro* and *in vivo* studies clearly indicate a large portion of transplanted cells remains differentiated into the desired cell type in the long-term.

COMMENT 1-2: *The statistical analysis used should be mentioned in the methods and significances indicated in the graphs.*

RESPONSE 1-1: Thanks for your kind suggestion! In the revised MS, we included the description of statistical analysis in the figure legends (**Fig. 1, Fig. 5-6, SI Fig. 5-6, 7, 9, 15, 22-25**). We have also added a description of statistical analysis in the corresponding method sections.

OTHER POINTS FOR FURTHER CONSIDERATION (FROM REVIEWER 1) INCLUDE:

COMMENT 2-1: *Abbreviations should be introduced at their first use and then used throughout the text / figures (e.g. weeks post injury - wpi).*

RESPONSE 2-1: Thanks again for your kind comment! Now the abbreviations (WPI – Line 281) are introduced at their first appearance and then used throughout the text/figures, correspondingly. Other abbreviations were also explained (Syn1 and MAP2 – Line 202; DAPT – Line 246; GFAP – Line 261; GFP – Line 316).

COMMENT 2-2: *Indicate the time point of sacrifice clearly in the figure legend (e.g. Fig. SI 21).*

RESPONSE 2-2: Time points have been now included in **SI Fig. 18-25** in their figure legends.

COMMENT 2-3: *How long were the iPSC-NSC cultured for? FGF2 can inhibit oligodendroglial differentiation and early passages produce more neurons compared to later ones. "Similar to the rNSC protocol" – this part was taken out and should have not be referred to.*

RESPONSE 2-3: iPSC-NSCs with passage 8-11 were used in all our transplantation. This description has now been added into the methods part. Same passage of iPSC-NSCs was used when comparing different substrate (glass, graphene oxide, and MnO₂ nanoscaffold) for *in vitro* neuronal differentiation studies. Passage number of iPSC-NSCs was also kept the same (8-11) when used for comparing different transplantation methods (injection, PCL and MnO₂ nanoscaffolds). Therefore, while higher passage number could produce more neurons, the results within each *in vitro* and *in vivo* experiment are comparable between different conditions. We also removed the sentence of “similar to the rNSC protocol” and replaced it with a direct description on the iPSC-NSC culture protocol.

COMMENT 2-4: *It should be mentioned in the figure legend of Fig. 4C that samples were incubated for 2 days.*

RESPONSE 2-4: Now we have included the description on time points in the figure legend of **Fig. 4**.

COMMENT 2-5: *The sentence in lines 204-206 needs to be fixed.*

RESPONSE 2-5: The redundant sentence now has been deleted.

RESPONSE TO SPECIFIC COMMENTS MADE BY REVIEWER 2:

COMMENT: *Thank you for thoroughly addressing the previous comments. However, a few concerns remain.*

RESPONSE: We appreciate the reviewer for his/her constructive comments and support. Now by revising the manuscript and supplementing new data, we addressed these concerns.

COMMENT 1-1: *The statistical methods used to assess significance must be included in the text – preferably in the appropriate figure captions and supplementary methods.*

RESPONSE 1-1: To follow up the comment, we included the description of statistical analysis in the methods and significances indicated in the graphs (**Fig. 1, Fig. 5-6, SI Fig. 5-6, 7, 9, 15, 22-25**). We have also added descriptions of statistical analysis in the corresponding method sections.

COMMENT 1-2: *In a few places (e.g., Introduction lines 77, 78), the authors argue that redox-tunable biodegradable is an advantage as stem cell scaffolds for SCI therapy. Although it is true that this property should make the scaffolds readily degrade in an inflammatory environment, it is not clear that this is an advantage for clinical applications – where inflammation may vary widely – or for delivery of neurogenic compounds such as DAPT – which ideally would be delivered over longer times to affect stem cell differentiation. Similarly, how would degradation of these materials be controlled clinically using scaffold thickness? It seems like a altering scaffold size is not ideal for clinical settings – where patient size and injury type may determine what scaffold size is required.*

RESPONSE 1-2: We thank the reviewer for bringing out this point. While having a redox-based biodegradation is unique, having tunable biodegradation is what makes our scaffold advantageous. To avoid this confusion, we deleted the word of “redox-” in the claimed sentence. We showed reliable control over scaffold degradation not only through changing scaffold sizes but also by introducing protein as a spacer, which effectively modulates porosities of scaffolds (**SI Fig. 10**). We also demonstrated the control of degradation by simply varying the amounts of cells transplanted without a need of any exogenous redox or inflammatory microenvironment. Therefore, we have several ways other than controlling scaffold size and geometry for tuning degradation, which is more relevant in clinical settings. It is not so practical to claim a specific degradation and drug delivery rate to be ideal across different patients or across species; however, the ability to control degradation speed in a wide range (from one day to over one month) by our nanoscaffold can be beneficial for on-demand drug delivery and personalized disease treatment.

COMMENT 1-3: *There are a few instances where the authors described BHI scaffolds as having “enhanced binding affinity toward ECM proteins” (e.g., line 103). Enhanced protein adsorption compared to what?*

RESPONSE 1-3: When we mention “enhanced binding affinity towards ECM proteins”, we are comparing to glass substrates and conventional polymer scaffolds. Now we have updated this sentence in Line 106 to improve its clarity.

COMMENT 1-4: The data show glass as a negative control, which proteins often absorb poorly to and is not therapeutically relevant. For laminin coating to be effective, glass is typically first etched and then often coated with polylysine before adding laminin. It does not appear like this was done for laminin-coated glass controls here and thus it is difficult to discern whether changes in the conformation of laminin adsorbed to glass versus nanosheets or simply the concentration of laminin is responsible for differences in cell adhesion, spreading and differentiation.

RESPONSE 1-4: To address this concern, we performed new laminin binding assay on chemically and plasma-etched, polylysine-coated glass substrates as well as on a polymer nanoscaffold [polycaprolactone (PCL) nanofiber that has been widely used in neural tissue engineering as well as used for our *in vivo* studies as a control group].

Experimental conditions were kept the same for all the experiments in BCA assay and each condition was repeated three times to obtain statistical information. As can be shown in updated Fig. 1c and consistent with previous experiments, both etched glass and polymer substrates show significantly lower binding towards laminin as compared to the experimental condition (MnO₂). Now we have updated Fig. 1 with the new data. From this new experiment, we could more reliably confirm the higher laminin density on our nanoscaffold is responsible for differences in cell adhesion, spreading and differentiation.

Updated Fig. 1c. A significantly upregulated ECM protein (laminin) binding towards 2D MnO₂ nanosheet, compared to control substrate [etched glass and polycaprolactone (PCL) substrates). These upregulated laminin binding was studied by a BCA protein assay. Data are mean±s.d. n=3, **P<0.01 by one-way ANOVA with Tukey post-hoc test.

COMMENT 1-4: On a similar note, how does adsorption to MnO₂ nanosheets affect laminin structure? Many of the sites in laminin known to bind cell receptors rich in aromatic and amine groups (e.g., IKVAV and YIGSR).

RESPONSE 1-4: The adsorption of laminin to MnO₂ nanosheet is through intermolecular interactions existent in many sites of laminin protein, and we also believe that some cell receptor binding sites (α1β1, α2β1, α3β1, α6β1,α7β1) on laminin would also interact with MnO₂ nanosheets. While an enhanced binding between MnO₂ and laminin can increase laminin intensity, MnO₂ binding towards cell receptor binding sites can counteract the upregulated focal adhesion. To investigate the effects on focal adhesion, we performed gene analysis on focal adhesion-related pathways (FAK and GAP43) that are associated with neural stem cell adhesion and differentiation. The significant upregulated FAK and GAP43 genes indicate a positive role of MnO₂ on the focal adhesion despite its possible interaction with certain cell-binding receptors.

SI Fig. 6f. Upregulated focal adhesion on MnO₂ nanoscaffolds. Data represents mean±s.d., n=3, ***P<0.001 by unpaired student t-test.

COMMENT 1-4: Finally, what type of laminin was used (was it laminin I or another form?)

RESPONSE 1-4: We used natural mouse laminin (Thermo Fisher, Catlog No.: 23017015), which is in the form of Laminin 1 (EHS laminin). Now we have included this description in the methods (A.8).

COMMENT 1-5: *The authors should be more cautious when claiming that the transplanted cells really become “functional” neurons after only 7 weeks in vivo. While they may have some electrical activity (Ca²⁺ imaging) and immature synapses (synapsin protein) there is no evidence that these cells have integrated (or are capable of integrating) with functional circuits. Even the BMS data showing functional recovery could easily be due to production of anti-inflammatory and regenerative factors by the stem cells, rather than their incorporation into host circuitry.*

RESPONSE 1-5: Thank you for your suggestions to improve manuscript clarity. We removed the word of “functional” and used the word of “mature” instead (updated in Line 201 as well as in the figure legend), as we used mature neuronal marker MAP2 and Synapsin for the immunostaining.

COMMENT 1-6: *For the in vivo data shown in Figure 5, the captions states that “Error bars are the standard error of the mean (n=2)”, implying that 2 separate animal cohorts were evaluated. However, how many animals were included in each cohort? Similarly, for counts of GFP+ cells (as in SI Fig. 20), was one section counted per region (SI Fig. 20f)? For Figure 6, the legend says that only 3 sections were counted per animal. Were these all in the same region (e.g., center of the scaffold implant)? What about for other in vivo quantification data (e.g., Caspase3, PH3)? Also, how were cells counted? Manually or by stereology? This method should be described.*

RESPONSE 1-6: In the BMS score (**Fig. 5**), two animals were used in each group and 2 individual BMS scores from different observers were given to each animal at each time point. Therefore, in total four scores were recorded at each time point for each individual experimental and control group. To avoid the confusion, we described the data collection more clearly in the method section and we supplemented statistical analysis as well.

In **SI Fig. 20**, when we count GFP+ cells, we used the automatic detection function in the NIS Nikon software to identify GFP+ cells shown in **SI Fig. 20e**, then summarized the amount of GFP+ cells at specific distance intervals (100 μm) in the sagittal sections. Therefore, they are not counted as individual transverse sections as shown in **SI Fig. 20f**. We used sagittal section in **SI Fig. 20e** and transverse section in **SI Fig. 20f** to give different information to the readers.

In **Fig. 6** and **SI Fig. 19-24** (including TuJ1, GFAP, Caspase 3 and PH3), 3 sections were counted per animal. We selected all these sections in the center of the scaffold implants and then identified immediately adjacent to the transplanted sites. The selection procedure was kept the same for the experimental condition (3D BHI nanoscaffolds) and control groups.

To quantify *in vivo* analysis data, all cells (GFP+, TuJ1+, MAP2+ cells) were first identified by automatic detection function in the NIS Nikon software (NIS element AR) then the number of cells were recorded for making the graphs. We have added this description in the methods part (A.7, A.18) to avoid confusion.

COMMENT 1-7: *Since biodegradation of scaffolds was assessed by scaffold thickness (line 165), does this imply that scaffolds degrade by surface erosion and not bulk degradation? If this is the case, do nutrients/water for embedded cells get in between the nanosheets, just not through the nanosheets?*

RESPONSE 1-7: We appreciate the reviewer’s discussion. We have assessed scaffold degradation by both scaffold thickness (**SI Fig. 9**) and by elemental analysis (**SI Fig. 10**). We did not investigate the mechanism of scaffold degradation in detail yet, but surface erosion could be a possible mechanism. In the case of surface erosion, as the scaffold stacked from nanosheets is porous, biodegradation cues can penetrate inside the scaffold and trigger the erosion inside the scaffold as well, thereby providing porosities and nutrients inside. Additionally,

embedded cells by themselves without any additional degradation/erosion cues can also degrade the scaffold and access the nutrients. Meanwhile, as our nanoscaffold is assembled from nanosheets and cells, nanoporosities exist throughout the nanoscaffold. Therefore, nutrients/water for embedded cells could also get in between the porosities of nanosheets other than completely relying on the degradation.

COMMENT 1-8: In the Summary (line 348), the authors state that our developed biodegradable hybrid inorganic nanoscaffold-based stem cell therapeutic approach would be a useful tool for selectively controlling stem cell differentiation and neuronal behaviors *in vivo*". While it may be able to do this with future refinements, in the current form the data only show the good survival of transplanted cells and the capacity of these cells to become neurons. The data do not show that differentiation and behavior can be selectively controlled.

RESPONSE 1-8: Based on this comment, now we have refined the sentence into "a useful tool for improving stem cell survival and inducing neuronal differentiation *in vivo*" (Line 356).

Minor comments from reviewer 2:

COMMENT 2-1: In the Methods, the authors describe using GFP-encoding lentivirus to "transfect" iPSC-NSCs prior to transplanted. However, the correct term would be "transduce" or "infect". "Transfect" would be used to describe plasmid delivery.

RESPONSE 2-1: We thank the reviewer's suggestion on using more proper terms. Now we have updated this description in the method part (A.18).

COMMENT 2-2: The images in Fig. 2e are difficult to see.

RESPONSE 2-2: In Fig. 2e, the image before degradation is hard to see because the scaffold is dark-colored before degradation while cell layers are semi-transparent. Therefore, only after the cleavage of the dark-colored scaffold, light-colored cell layers could be seen. To better clarify and explain the figure, we included a description in the figure legend.

COMMENT 2-3: Abstract (line 49) – The wording of this sentence ("To this end, we developed a novel biodegradable nanoscaffold-based method for the controlled delivery of therapeutic molecules, and improved spinal cord injury (SCI) treatment through enhanced stem cell therapy by improving stem cell survival, neuronal differentiation, and neurite outgrowth in a mouse model.") is oddly redundant and really not clear.

RESPONSE 2-3: Now we have modified this sentence as "To this end, we developed a novel biodegradable nanoscaffold-based method for enhanced stem cell therapy, drug delivery and treatment of spinal cord injury (SCI)" (Line 47-49).

COMMENT 2-4: In the Introduction (line 63), the authors broadly refer to "stem cell therapy". However, this should read "stem cell transplants" in the context of the rest of the sentence so that therapies aimed at targeting endogenous stem cells are excluded.

RESPONSE 2-4: Now we have switched it into "stem cell transplantations" (Line 59) to avoid confusion.

COMMENT 2-5: In the Introduction (line 66), please include a citation to support the claim that most cells die shortly after transplantation.

RESPONSE 2-5: Now one representative reference has been added (Line 65)³. Original sentence in the cited paper: “However, in many cases, large numbers of transplanted cells die after transplantation...”. To improve the precision of this sentence, we have also changed the word “most” to “many”.

COMMENT 2-6: *Introduction (line 68) – “Therefore” can be deleted.*

RESPONSE 2-6: Now it has been deleted (Line 66).

COMMENT 2-7: *Introduction (lines 80-81) – The authors state that BHI scaffolds “have extraordinary properties” but do not elaborate on these in this section so that the sentence reads like “filler” with no real content.*

RESPONSE 2-7: As this sentence does not provide much information, now we have removed this sentence to improve the flow (Line 78-79).

COMMENT 2-8: *Introduction (line 88) – “applications” should be “application”*

RESPONSE 2-8: Now we have corrected this grammar error (Line 86).

COMMENT 2-9: *Lines 294-297 – It seems that stem cell survival may reduce glial scar formation (through production of anti-inflammatory factors) and stem cell differentiation may help re-establish neuronal circuits. In its current form, the text implies that both cell survival and differentiation may prevent glial scar formation, which has not really been shown.*

RESPONSE 2-9: To correct this confusing claim, now we have updated the sentence from “which results” to “and can result” (Line 300).

REFERENCES FOR THIS RESPONSE LETTER:

- 1 Su, Z., Niu, W., Liu, M.-L., Zou, Y. & Zhang, C.-L. In vivo conversion of astrocytes to neurons in the injured adult spinal cord. *Nature communications* **5**, 3338, (2014).
- 2 Assinck, P., Duncan, G. J., Hilton, B. J., Plemel, J. R. & Tetzlaff, W. Cell transplantation therapy for spinal cord injury. *Nature Neuroscience* **20**, 637, (2017).
- 3 Thuret, S., Moon, L. D. & Gage, F. H. Therapeutic interventions after spinal cord injury. *Nature Reviews Neuroscience* **7**, 628, (2006).

Reviewers' comments:

Reviewer #1 (Remarks to the Author):

Revision II:

I appreciate the authors effort to clarify my concerns with the manuscript; however, some aspect have not been well answered or rather the data is not very compelling, and while the clarity improved, it was also made clear that some conclusion cannot be drawn from the data collected due to an insufficient amount of animals. More specifically:

(I) Differentiation of human cells takes a long time, a one month time point is not sufficient to investigate the long term cell fate (see e.g. doi: 10.1172/JCI92955). In addition, the percentage of neurons significantly drops compared to their earlier time point, which indicates that the long term promotion of neuronal differentiation / survival is limited. There is only a minor (if any) increase in neuronal differentiation with their scaffold compared to other methods at similar or even later time points. Although not plotted, it looks like there is no difference anymore between their scaffold and the injected group (control) at the one month time point in terms of TuJ1+ cell percentage (fig. SI25 b). It is unclear to me why the authors did not quantify their seven week time point? In addition, the survival seems very low at the seven week time point.

(II) The authors indicate that 2 animals per group were used for the behavioural test, which is not sufficient to draw any conclusion. The practice to count 2 BMS scores from two observers as 4 scores is questionable at best, but I appreciate that the authors were honest in how they acquired these scores.

Taken together, the study only shows limited long term survival and limited, if any, improvement in neuronal differentiation, coupled with an underpowered behavioural study.

Reviewer #2 (Remarks to the Author):

Thank you for thoroughly addressing comments in the previous review. It looks great except for one minor point. On line 170, please specify that tunability of biodegradation rate was achieved by changing the number of assembled layers.